# *Trypanosoma brucei* cattle infections contain cryptic transmission-adapted bloodstream forms at low parasitaemia

Stephen D. Larcombe[1], Edith Paxton [2], Christina Vrettou[2], Pieter C. Steketee [2], Keith R. Matthews [1], Liam J. Morrison [2] & Emma M. Briggs [1,3] ✉

Tsetse-transmitted *Trypanosoma* parasites infect a wide host range and cause Human African Trypanosomiasis and Animal African Trypanosomosis. The dominant hosts of *Trypanosoma brucei sensu lato* are non-human mammals, including agriculturally important cattle. In rodent infections, *T. brucei* transitions from proliferative slender to tsetse-transmissible stumpy forms at high parasitaemia in a density-dependent quorum sensing-type process. However, chronic bovine infections are characterised by markedly lower blood parasitaemia levels; mostly substantially below the density assumed to trigger slender-to-stumpy differentiation. This challenges the current (rodent-based) assumptions and quantitative parameter estimations around stumpy form generation in the bloodstream. By combining scRNA-seq and microscopy we observe mixed populations of parasites with both slender- and stumpy-associated transcriptomes in cattle blood. The appearance of the latter coincides with fewer detectably dividing parasites and parasites with shortened flagellum indicative of differentiation, despite the absence of stumpy morphology or developmental marker protein expression. Comparisons with murine infections and in vitro culture demonstrates conserved transcriptomic signatures for both slender- and stumpy-like forms, as well as host specific differences, including a subpopulation of slender-like parasites upregulating pyruvate metabolism and TCA cycle transcripts in cattle samples. These similarities and differences are key to understanding parasite development and transmission in its natural host.

Tsetse-transmitted African trypanosome parasites are a significant public health and socio-economic problem across 37 African countries where tsetse flies are endemic, due to the parasite's ability to infect both humans and a wide range of domesticated and wild animals[1,2]. The best studied trypanosome species, *Trypanosoma brucei*, is the causative agent of Human African Trypanosomiasis (HAT; subspecies *T. b. gambiense* and *T. b. rhodesiense*), and is one of the causes of

African Animal Trypanosomosis (AAT or Nagana, *T. b. brucei*) in livestock. Thanks to extensive vector control efforts, human cases of trypanosomiasis are scarce relative to historic levels, but disease in cattle remains responsible for a substantial economic and humanitarian burden[3].

Most African trypanosome species depend on the tsetse fly vector to complete their life cycles, requiring these parasites to undergo a

[1]Institute for Immunology and Infection Research, School of Biological Sciences, Ashworth Laboratories, University of Edinburgh, Edinburgh, UK. [2]Roslin Institute, Royal (Dick) School of Veterinary Studies, University of Edinburgh, Midlothian, UK. [3]Newcastle University Biosciences Institute, Faculty of Medical Sciences, Newcastle University, Newcastle upon Tyne, UK. ✉e-mail: emma.briggs@newcastle.ac.uk

complex series of developmental changes when passing from a mammalian host to the insect vector[4]. Among African trypanosomes, *T. brucei* is unique in that its development to transmissible forms in mammalian hosts is marked by a distinct morphological change; replicative 'slender' forms establish the infection, before differentiating into non-dividing 'stumpy' forms in a density-dependent manner[5,6]. These stumpy forms are cell cycle arrested, relatively resistant to antibody-mediated complement lysis in comparison to slender forms[7], and exhibit molecular pre-adaptation for their onward development in tsetse flies[8]. Although slender forms can infect immunologically compromised and immature tsetse, stumpy forms preferentially infect adult tsetse flies, demonstrating their importance for disease transmission[9]. Indeed, parasites that have lost the capacity for stumpy formation (monomorphs) fail to complete development in the fly[10,11].

Most studies of slender and stumpy form dynamics in the mammal have employed rodent models of infection; here, parasitaemia levels in the blood undergo striking "peaks and troughs" in early infections, and parasitaemias can exceed $10^8$ parasites per ml, with infections often fatal within weeks[12]. Single-cell transcriptomics (scRNA-seq) revealed that most bloodstream parasites in these rodent infections, especially in the chronic phase when parasitaemia remains high, are either fully developed stumpy forms or differentiating intermediate forms[13]. Most intermediate forms are cell cycle arrested, express a stumpy-associated transcriptome profile, and are committed to stumpy-development, but are yet to develop the stumpy morphology[13]. Thus, morphology alone was not indicative of developmental status. Rodent models, however, are not reflective of cattle infections, or those of wild bovids and other species, which are characterised by low parasitaemia in the chronic phase of infection, and which can persist for months to years, especially under field conditions[14]. This impedes detailed longitudinal study of *T. brucei* in cattle when parasitaemia falls below the detection threshold (usually by microscopy). Indeed, in classic studies using infections of different hosts, parasites were often only detectable following sub-inoculation in rodents, or through fly infection (e.g. ref. [15]). Alternatively, PCR can be used to detect the presence of DNA deriving from particular species. This approach has revealed that *T. brucei* DNA can be identified in the skin and adipose tissues of cattle, goats and sheep[16], suggesting similarities to mouse models[17,18], where extravascular localisation in these tissues is well characterised. However, while the contribution of tissue-resident *T. brucei* to sustaining infection[13,19] and transmission[17] is clear in mice, the nature, location and scale of extravascular tissue invasion is unclear in cattle and other livestock.

Blood parasitaemia in chronic cattle infection persists well below the high density of parasites that is documented to produce morphologically stumpy forms in both rodent and in vitro models ($>10^6$ parasites per ml). Consequently, how stumpy forms arise and support transmission at low parasitaemia remains a conundrum, and parasite populations in bovine infections have not been characterised in detail or compared to those of murine or in vitro models —particularly in the context of chronic infection, or with the extensive molecular knowledge of differentiation that has been generated in the last decade for *T. brucei*.

In this study, we asked three questions: what are the molecular characteristics of *T. brucei* bloodstream forms during cattle infection; do any identified cell types fit our current understanding of slender-to-stumpy development, and, finally, do cattle parasites differ from bloodstream forms studied using in vitro and rodent models? To address these, we conducted a molecular investigation of the development of *T. brucei* in the cattle bloodstream, with emphasis on parasite dynamics after the first waves of parasitaemia. We followed *T. b. brucei* infections for 60 days in calves and combined quantitative morphological parameters with scRNA-seq to characterise individual parasites within the potentially heterogeneous trypanosome population dynamically over time. We found that both slender- and stumpy-like transcriptomic forms were present in varying proportions across infection, with stumpy-like forms reaching as much as 68% of the population in early infection. We also found few actively dividing parasites, consistent with low levels of proliferation leading to low blood parasitaemia. Nonetheless, we found a paucity of parasites with classical stumpy morphology, or stumpy protein marker expression, despite their stumpy-associated transcriptome. Instead, parasites had a morphology more closely resembling an intermediate/differentiating population. We also note host-specific differences between bloodstream forms either generated in vitro or isolated from mice or cattle, including a subpopulation in cattle samples with altered expression of TCA cycle components. By examining the developmental biology of trypanosomes in their natural hosts and in a phase of infection most representative of parasitaemias in the field, we provide new insight into how transmission to tsetse flies from cattle is supported.

## Results

### Parasitaemias in cattle comprise mixed populations expressing both slender- and stumpy-associated transcriptomes

Two calves were infected via the intravenous route with $1 \times 10^6$ *T. b. brucei* Antat 1.1 90:13, and blood parasitaemia was monitored over 60 days (Fig. 1A, Supplementary Fig. 1). Within the resolution of analysis, parasitaemia peaked between days 5 and 23 post infection, reaching an observed maximum of $4 \times 10^4$/ml and $3 \times 10^5$/ml in cows 1 and 2, respectively, before entering a chronic phase characterised by parasitaemia $<1 \times 10^4$/ml after day 23. In this chronic phase, for both cattle, parasitaemia persisted mostly between $1–3 \times 10^3$/ml, or occasionally fell below the threshold of detection. These parasitaemia dynamics and variability between calves are consistent with previous experimental infections of cattle with the same parasite strain[20].

To investigate the potentially heterogeneous populations of *T. brucei* in the cattle bloodstream, we used scRNA-seq at time points when sufficient parasites could be isolated and purified; days 4, 11 and 23. Parasites were also purified from the blood at the end of the experiment (day 60) although extremely low parasite numbers at this time point hampered efforts to obtain sufficient cells for Chromium scRNA-seq analysis. Despite this, we obtained between 1585 and 13,747 individual transcriptomes from *T. brucei* for both cows on days 4, 11, and 23, as well as 257 transcriptomes from cow 2 on day 60 (Supplementary Fig. 2). Transcriptomes obtained from cow 1 on day 60 did not meet quality control thresholds for analysis and were excluded.

Samples were integrated together before dimension reduction and clustering analysis (Fig. 1B). Four distinct clusters were identified, and these were detected in all samples with the exception of cow 2 day 60 (Fig. 1C) where the low number of transcriptomes captured limited analysis. Independent approaches identified that three of the clusters (0, 2 and 3) had a transcriptome consistent with slender forms but varied in other aspects, whereas cluster 1 (green) contained transcriptomes that more closely resembled stumpy forms. Firstly, the average expression level of previously identified slender-associated markers[21] was lower in cluster 1 compared to the other clusters and, conversely, the average expression of stumpy-associated markers was higher in cluster 1 compared to clusters 0, 2 and 3 (Fig. 1D). Secondly, de novo identification of marker genes associated with each cluster was conducted via differential expression analysis between the clusters (Supplementary data 1, Supplementary Fig. 3). The ten markers with the highest fold change were plotted for each cluster (Fig. 1E). This included procyclin encoding transcripts, the top markers of cluster 1. This was followed by gene ontology (GO) term enrichment to identify biological processes associated with each cluster (Supplementary Fig. 3B, Supplementary data 1), discussed in detail below. Thirdly, the cell cycle phase of each cell was classified using the expression of previously defined phase marker genes[22] (Fig. 1F). The average

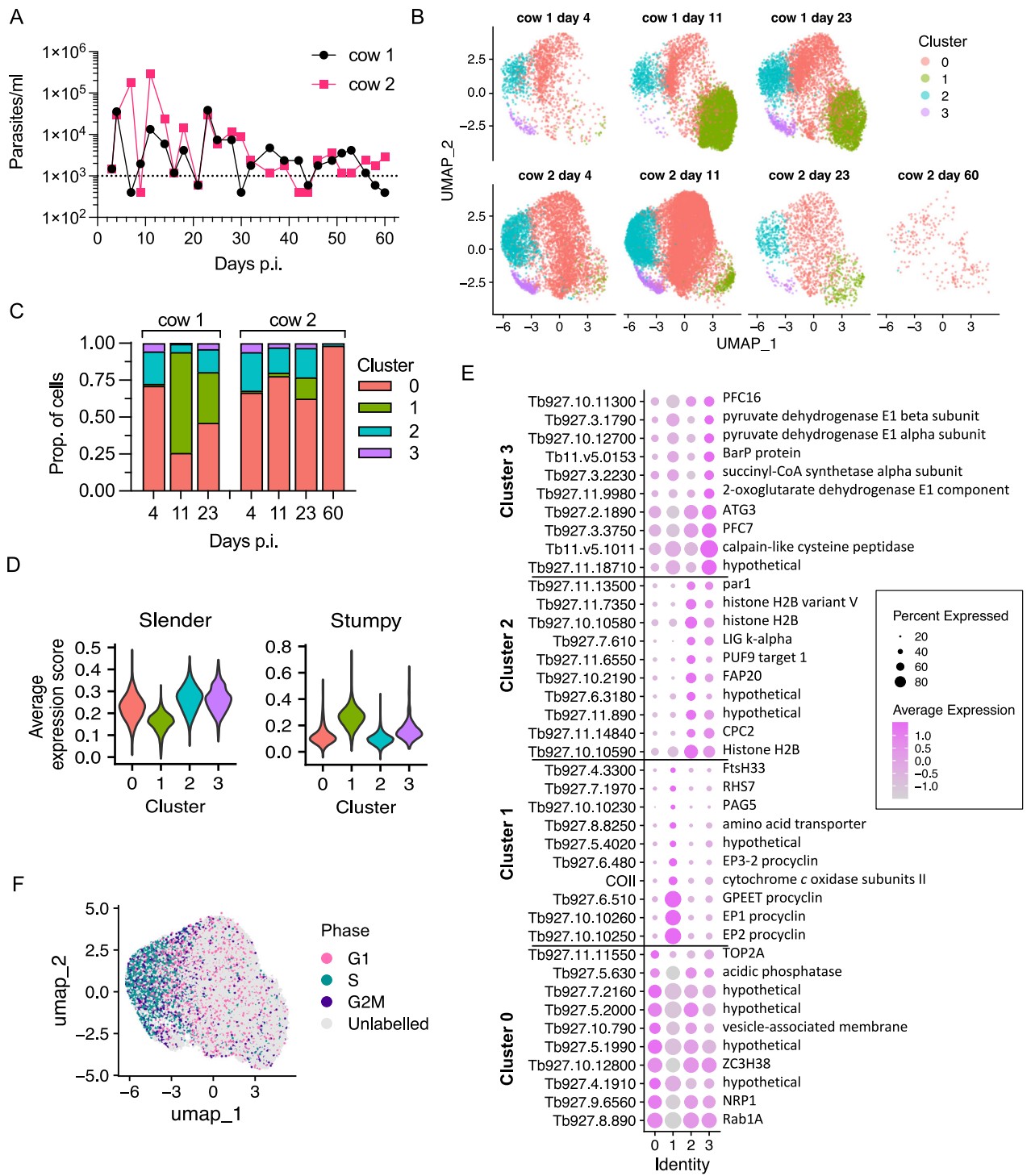

**Fig. 1 | scRNA-seq identified varied populations of *T. brucei* bloodstream forms in cattle. A** Blood parasitaemia levels for cow 1 (black) and cow 2 (pink). Dotted line indicates the low confidence threshold. **B** Clustered individual transcriptomes of *T. brucei* isolated on days 4, 11, 23 and 60 (only cow 2) post infection. Cluster colours are consistent across plots. **C** Proportion of each sample, split by cow, in each of the clusters. **D** Average expression scores of slender (left) and stumpy (right) marker genes for each cluster. Scores above 0 indicate higher average expression of the marker gene set identified previously[21], compared to a set of randomly select control genes. Cluster colours as in (**B**). **E** Dot plot of the average expression of the top ten marker genes for each cluster with the highest fold change (colour scale). The size of each dot is proportional to the percentage of cells in that cluster expressing the marker. Gene name or description is shown to the right. **F** UMAP of all samples coloured by cell cycle phase. "Unlabelled" indicates the cell did not overexpress any phase marker genes (G1, S or G2/M), compared to a random set of control genes. Source data are provided as a Source data file.

expression of G1, S and G2/M marker genes was calculated per cell and compared to the average expression of a set of random control genes to generate an "average expression score" for each phase (Supplementary Fig. 4A). Cells were labelled with the phase with the highest expression score, or "Unlabelled" if average expression was not above that of the control gene set for any phase.

Slender-like clusters 2 and 0 were largely distinguished by differences in their cell cycle phase proportions (Fig. 1F, Supplementary

Fig. 4B). Cluster 2 showed high proportions of S (46%) and G2/M (13%) phase labelled cells, consistent with the top marker genes (i.e. chromosome passenger complex 2 and histones, Fig. 1E) and associated GO terms that include "DNA packing" and "cytokinesis" (Supplementary Fig. 3B). Hence, cluster 2 consisted of proliferative slender forms. Cluster 0 in contrast was largely made up of "Unlabelled" cells (79%) and had the fewest unique markers (Supplementary Fig. 3A), with the top markers also detected in the other slender-like clusters 2 and 3 (Fig. 1E). GO terms for cluster 0 include "purine nucleoside disphosphate metabolic process" and "phosphorylation" due to expression of slender-related genes including hexokinase and pyruvate kinase 1 (Supplementary Fig. 3B, Supplementary data 1). This population hence appears to contain few parasites in S or G2/M phases, but retains expression of other slender-associated markers.

Cluster 3 contained cells in all cell cycle phases, with 17% and 15% residing in S and G2/M, respectively, but with a large proportion of unlabelled cells, indicating it was made up of both dividing and non-dividing parasites (Supplementary Fig. 4B). This cluster was distinguished by a distinct set of marker genes relating to "tricarboxylic acid cycle" and "pyruvate metabolic process" (Fig. 2A), including mitochondrial malate dehydrogenase and 2-oxoglutarate dehydrogenase (Fig. 2B). This subpopulation of slender-like forms had upregulated the expression of genes required to further metabolise pyruvate to generate ATP (Fig. 2B, C), distinct from the classically described secretion of pyruvate by slender forms[23], but maintained and in some cases upregulated expression of glycolytic components (Fig. 2C).

In contrast, cluster 1 was associated with "translation" (Fig. 3A) due to the high number of ribosomal protein transcripts that showed elevated levels in this cluster, as well as eukaryotic translation initiation factor 5 A (EIF5A) and receptor for activated C kinase 1 (RACK1)[24]. Notably, previous transcriptomic analysis of *T. b. rhodesiense* isolated at peak parasitaemia in rats, when stumpy forms are dominant, also showed high levels of translation associated transcripts[25]. Cluster 1 markers (Fig. 3B, Supplementary data 1) also included: Protein Associated with Differentiation (PAD) family members that are involved in the perception of the stumpy-to-procyclic differentiation signal[26]; procyclic form surface proteins EP and GPEET procyclin[27]; genes encoded on the maxi circle kDNA, including a mitochondrial NADH dehydrogenase subunit ND4 and cytochrome oxidase subunit II; the stumpy-elevated and heat stress associated RNA regulator ZC3H11[28]; the iron responsive regulator RBP5 (Tb927.11.12100)[29]; Factor H receptor[30]; and, the pteridine transporter Tb927.10.9080, which is a target of snoGRUMPY, a proposed regulator of stumpy formation[31]. Therefore, cluster 1 parasites expressed known transcript markers of stumpy forms, and were strongly associated with increased expression of genes required for global translation and developmental gene regulation. This subpopulation is henceforth referred to as stumpy-like for these reasons.

## The appearance of parasites expressing stumpy-associated transcripts correlates with reduced proliferation and shorter flagella lengths

When possible, parasites were also examined by microscopy. Given the appearance of a subpopulation expressing stumpy-associated transcripts and the low number of parasites available, we prioritised analysing hallmarks of stumpy form development: cell division cycle changes, PAD1 surface protein expression, and changed morphology via reduced flagellum length (Fig. 4).

Dividing parasites first segregate the mitochondrial kDNA (K) followed by the nuclear genome (N) allowing cell cycle status to be inferred from the KN configuration. Parasites with segregated kDNA and 1 nucleus, 2K1N, represent those in late S phase or G2, whereas parasites with segregated kDNA and 2 nuclei, 2K2N, are post-mitotic. Proliferative parasites with 1 kinetoplast and 1 nucleus (1K1N) can be in G1 or early S phase, whereas stumpy cells have exited the cell cycle and uniformly

exhibit 1K1N. The parasites isolated from cow 2 on day 4, as the parasitaemia was rapidly ascending towards its peak in the first wave of infection, were enriched for dividing forms (43% 2K1N or 2K2N, Fig. 4B). All other samples profiled contained 2K1N/2K2N parasites, albeit in low numbers, indicating that the cattle blood contained dividing *T. brucei* at each point but as a very small proportion of the overall population relative to the establishment phase. Notably, the proportion of 1K1N parasites reached 96% and 98% in cow 1 on days 11 and 23, respectively, indicating low levels of division consistent with stumpy development[13]. These samples also contained the greatest proportions of cluster 1 parasites identified by scRNAseq analysis (Fig. 1C).

Further attempts to quantify the proportion of proliferative parasites were made using an ex vivo plating assay, which previously allowed accurate detection of replication-competent parasites when present in as few as 0.1% of the total population[13]. Parasites were seeded into 96-well plates at known dilutions (50, 20, 5, 2 and 1 parasite(s)/ well), with twelve wells prepared per dilution. Surprisingly, no samples gave rise to proliferating cultures with the exception of parasites isolated from cow 2 on day 4, in which 5/12 wells grew at 20 parasites/well seeding (estimated 3% viability based on probability equation, see methods), 4/12 wells at 5 parasites/well (11% viability) and 2/12 wells at 2 parasites/well (9% viability). Thus, it was estimated only 7.7% of parasites isolated from cow 2 on day 4 were viable and able to replicate ex vivo despite the large proportion of actively dividing cells (as defined by their 2K1N or 2K2N status). Given the detection of active cell division markers by both microscopy and scRNA-seq in all cattle samples, we consider that rapid adaptation to the bovine environment rendered the parasites unable to proliferate in culture media once isolated, regardless of their replicative capacity in vivo.

As the proportion of stumpy-like cluster 1 parasites varied most between the two cows on day 11 (Fig. 1C), we used this time point to correlate two established differentiation parameters with the appearance of this cluster: stumpy-specific protein PAD1 expression[26] and flagellar shortening that marks both intermediate and stumpy stages[32]. No parasites showed expression of PAD1 across the parasite surface in either sample. Note that PAD1 antiserum can bind the flagella of parasites irrespective of their differentiation status, and this includes monomorphic strain Lister 427 (Supplementary Figs. 5 and 6). In mouse infections, flagellar length provided an indication of morphological differentiation, being shorter in stumpy forms, but there was also a quantifiable change between slender forms from proliferating populations (mean = 27 μm) and differentiating populations (mean = 24 μm) that were yet to take on a full stumpy morphology (mean = 19 μm)[32]. Visual inspection of live parasites from cattle revealed that most parasites had a slender to intermediate morphology (see Supplementary video 1). Flagella length was used to quantify this morphological change and this was compared to the same parasite stabilate undergoing slender-to-stumpy morphological development in mice, thereby calibrating this transition in a bovine infection with events occurring in the well characterised murine model (Fig. 4C). The shortening of the flagellum was evident during mouse infection, as expected, as the parasites transitioned from a proliferative slender population (day 3, mean length 27.3 μm) to a differentiating population of non-proliferating forms that did not yet resemble full stumpy forms (day 5, mean length 23.9 μm), a heterogeneous population of PAD1 negative differentiating forms (day 6, mean 24.9 μm) and PAD1 positive stumpy forms (day 6, mean 21 μm), before all cells were PAD1 positive stumpy cells on d7 (mean = 21 μm). Notably, most parasites isolated from both cattle on day 11 had a reduced flagellar length (means = 22.3 μm and 24.4 μm, respectively), more consistent with differentiating forms than proliferating slender forms. Additionally, the flagellum length distribution differed between cow 1 and cow 2 on day 11, with more parasites with shorter flagella in cow 1, consistent with a more differentiated population. In contrast in cow 2, with more proliferative parasites, flagella were longer. This correlates with the

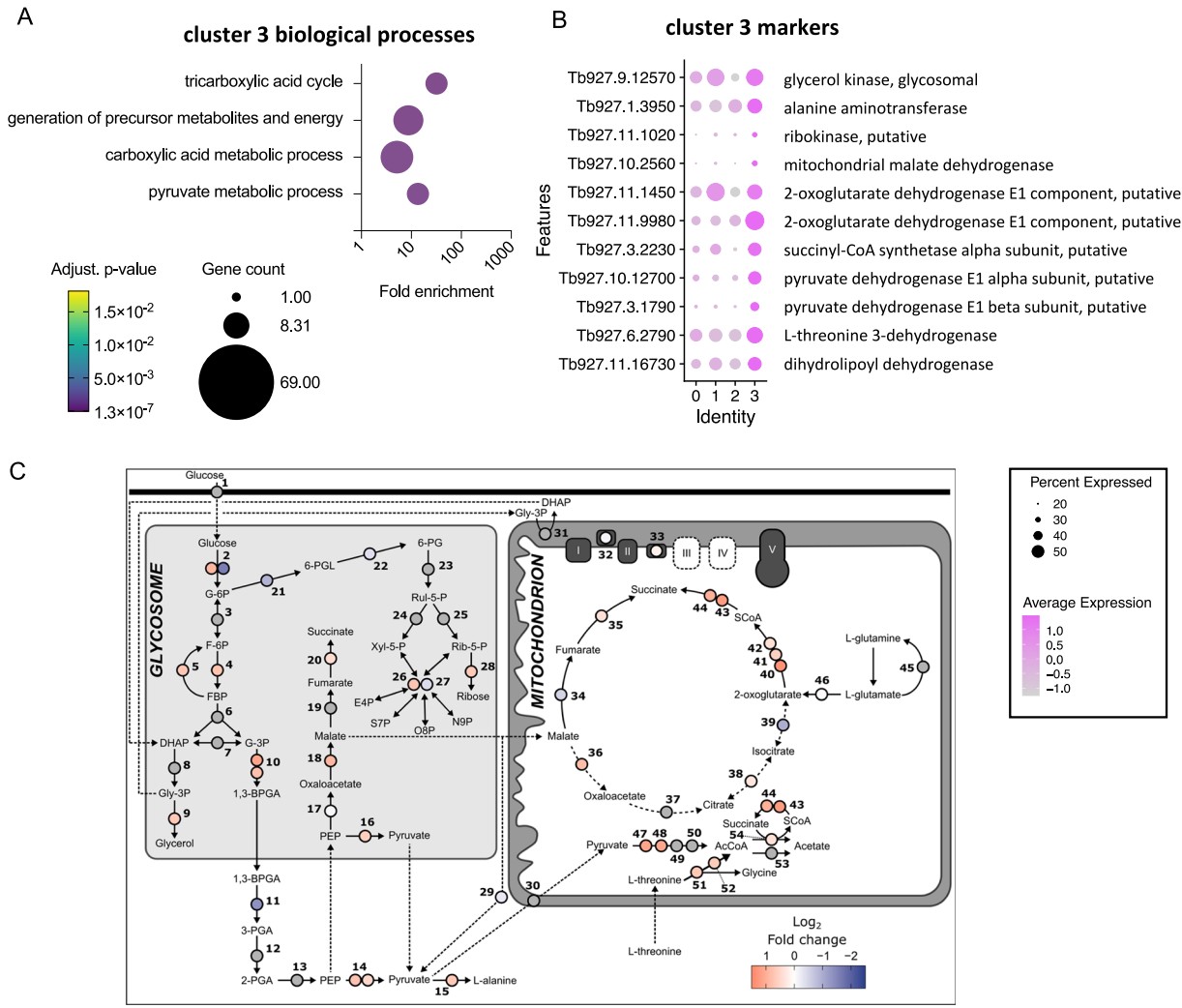

**Fig. 2 | A proportion of parasites in the cattle bloodstream have higher transcript levels of pyruvate metabolism linked genes. A** Biological process GO terms enriched for cluster 3 marker genes. Plot indicated the fold enrichment (x-axis), number of genes for each term (size) and the adjusted *p*-value (colour scheme) for each term (Benjamini-Hochberg adjusted *p*-values from Fisher's exact test). **B** Dot plot of the average expression of selected cluster 3 markers genes linked to terms "pyruvate metabolic process" and the "TCA cycle". **C** Log2 fold change (Cluster 3 compared to clusters 0, 1 and 2 combined) was calculated for each gene. Dashed lines represent transport processes. Grey circles indicate genes were not significantly different in cluster 3 relative to other clusters. Genes: 1, hexose transporters; 2, hexokinase, Tb927.10.2010 and Tb11.v5.0732; 3, glucose-6-phosphate isomerase; 4, phosphofructokinase, Tb927.3.3270; 5, fructose-1,6-bisphosphatase, Tb927.9.8720; 6, aldolase; 7, triosephosphate isomerase; 8, glycerol-3-phosphate dehydrogenase; 9, glycerol kinase, Tb927.9.12570; 10, glyceraldehyde 3-phosphate dehydrogenase, Tb927.6.4280 and Tb927.6.4300; 11, phosphoglycerate kinase, Tb927.1.710; 12, phosphoglycerate mutase; 13, enolase; 14, pyruvate kinase 1, Tb927.10.14140 and Tb11.v5.0605; 15, alanine aminotransferase, Tb927.1.3950; 16, pyruvate phosphate dikinase, Tb927.11.6280; 17, Phosphoenolpyruvate carboxykinase, Tb927.2.4210; 18, glycosomal malate dehydrogenase, Tb927.10.15410; 19, glycosomal fumarate hydratase; 20, glycosomal NADH-dependent fumarate reductase, Tb927.5.930; 21, glucose-6-phosphate

dehydrogenase, Tb927.10.2490; 22, 6-phosphogluconolactonase, Tb927.11.6330; 23, 6-phosphogluconate dehydrogenase; 24, ribulose-5-phosphate epimerase; 25, ribose 5-phosphate isomerase; 26, transketolase, Tb927.8.6170; 27, transaldolase, Tb927.8.5600; 28, ribokinase, Tb927.11.1020; 29, malic enzyme, Tb927.11.5450; 30, Mitochondrial pyruvate carrier 2; 31, FAD-dependent glycerol-3-phosphate dehydrogenase; 32, NADH dehydrogenase (NDH2), Tb927.10.9440; 33, Alternative oxidase, Tb927.10.7090; 34, mitochondrial fumarate hydratase, Tb927.11.5050; 35, mitochondrial NADH-dependent fumarate reductase, Tb927.10.3650; 36, mitochondrial malate dehydrogenase, Tb927.10.2560; 37, citrate synthase; 38, aconitase,Tb927.10.14000; 39, isocitrate dehydrogenase, Tb927.8.3690; 40, 2-oxoglutarate dehydrogenase E1 component, Tb927.11.9980; 41, 2-oxoglutarate dehydrogenase E1 component, Tb927.11.1450; 42, 2-oxoglutarate dehydrogenase E2 component, Tb927.11.11680; 43, succinyl-CoA synthetase α, Tb927.3.2230; 44, succinyl-CoA ligase β, Tb927.3.2230; 45, glutamine synthetase; 46, glutamate dehydrogenase, Tb927.9.5900; 47, pyruvate dehydrogenase E1 α subunit, Tb927.10.12700; 48, pyruvate dehydrogenase E1 β subunit, Tb927.3.1790; 49, dihydrolipoamide acetyltransferase; 50, pyruvate dehydrogenase complex E3; 51, L-threonine 3-dehydrogenase, Tb927.6.2790; 52, 2-amino-3-ketobutyrate coenzyme A ligase, Tb927.8.6060; 53, Acetyl-CoA hydrolase (ACH), Tb927.3.4260; 54, Succinyl-CoA:3-ketoacid coenzyme A transferase (ASCT), Tb927.8.6060. Source data are provided as a Source data file.

higher proportion of cells belonging to stumpy-like cluster C1 and with a 1K1N configuration in cow 1 than in cow 2 (Fig. 4D).

Thus, the short flagellum, prevalence of cluster 1 cells, and those with a 1K1N profile, each are indicative of a prevalence of stumpy-like parasites in cattle blood that, at least in the samples investigated, did not express PAD1 protein.

## In chronic infections, parasite morphology revealed mixed populations of both dividing and differentiating forms
Having established the cytological characteristics of the stumpy-like forms (based on their transcriptome), we used these features to investigate the bloodstream forms present in chronic cattle infection when parasitaemia levels dropped to around

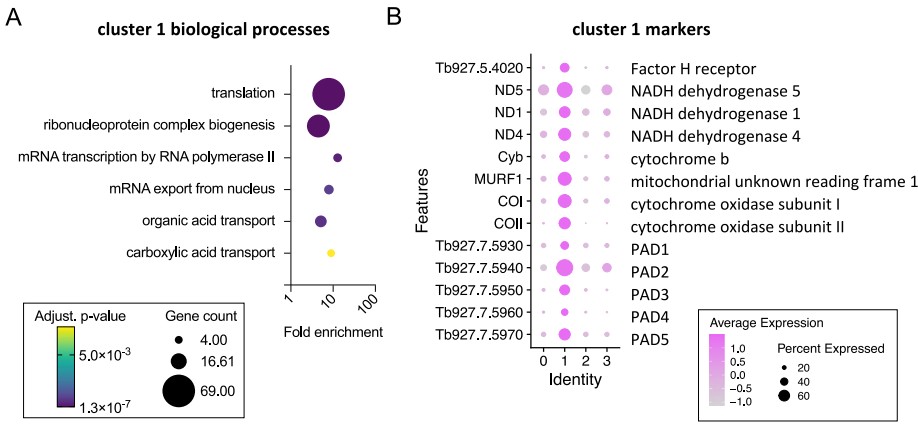

**Fig. 3 | A proportion of parasites in the cattle bloodstream having higher transcript levels of stumpy form associated genes. A** Biological process GO terms enriches for cluster 1 marker genes. Plots indicate fold enrichment (x-axis), number of genes for each term (size) and the adjusted *p*-value (colour scheme) for each term (Benjamini-Hochberg adjusted *p*-values from Fisher's exact test). **B** Dot plot of the average expression of cluster 1 marker genes previously linked to stumpy forms.

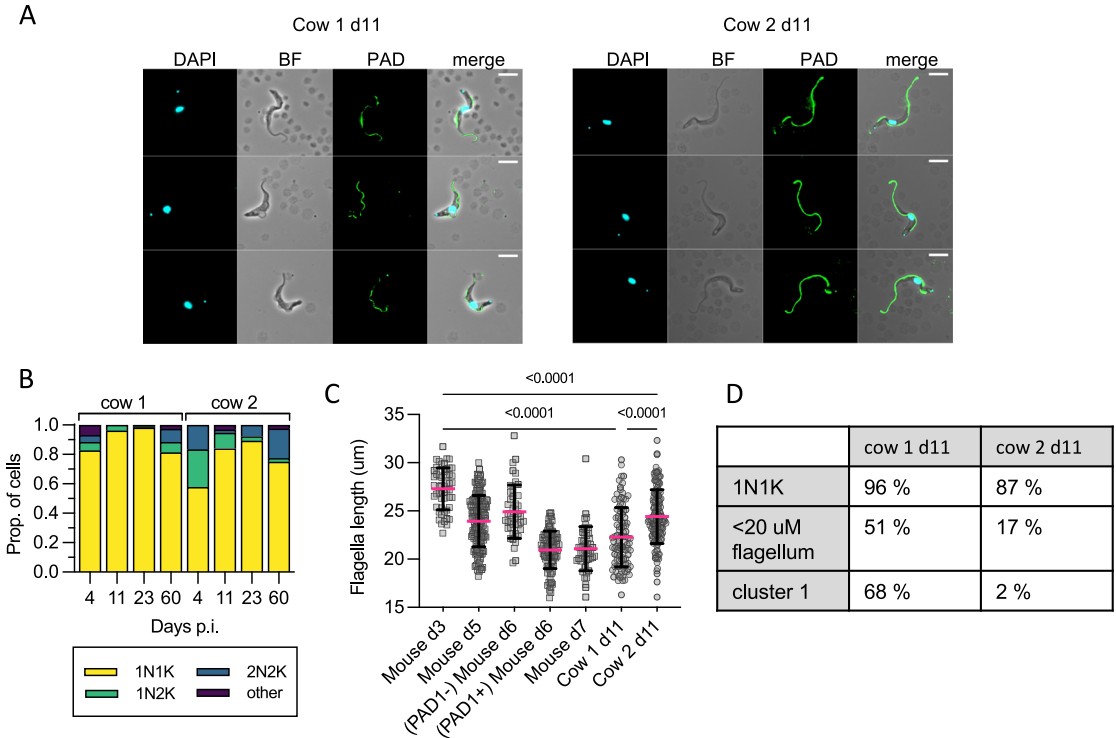

**Fig. 4 | Cell morphology of cattle derived bloodstream form *T. b. brucei*. A** Representative cell images of parasites isolated on d11 p.i. in cow 1 and cow 2, showing an absence of cell surface PAD1 staining (except the flagella). **B** KN counts of parasites in each cow at each time point. **C** The distribution of flagellar measurements from parasites in both cows on d11 p.i. compared to controls taken from mice. On d3 (*n* = 1, on d3 p.i. blood parasites are absent in most mice), all mouse parasites are slender, PAD1 negative, and fully replicative; on d5 (*n* = 3) all appear slender, are PAD1 negative, and only some are replicative, on d6 (*n* = 3) some parasites are PAD1 positive, very few are replicative; on d7 (*n* = 3) all are PAD1 positive and non-replicative. Parasites from the cows on d11 were significantly different from d3 true slender forms from mice, and from each other. *P*-values indicated show the result of two-sided Dunn's multiple comparison tests between samples. Data are presented as mean +/− SD. **D** A comparison table of the proportion of cells belonging to stumpy-like cluster 1, the proportion having a flagellar length most similar to mouse derived stumpy forms (< 22 μm) and number of non-dividing (1K1N) cells, showing the difference between cow 1 and cow 2 on d11 p.i. Source data are provided as a Source data file.

$1 \times 10^3$–$1 \times 10^4$ parasites/ml, below the level feasible for effective scRNA-seq analysis.

In this chronic stage, parasites were overwhelmingly 1K1N, particularly between days 49–58 (mean = 93.4% 1K1N, min = 89.4, max = 97.6; Fig. 5A, B). This was also observed when the parasitaemia was extremely low, indicating low levels of active parasite division in the blood during most of the chronic phase of infection. Despite the majority of parasites being non-dividing forms, only 2/824 parasites were observed that stained positively for PAD1, both from cow 2 on d58, which was consistent with the general absence of fully

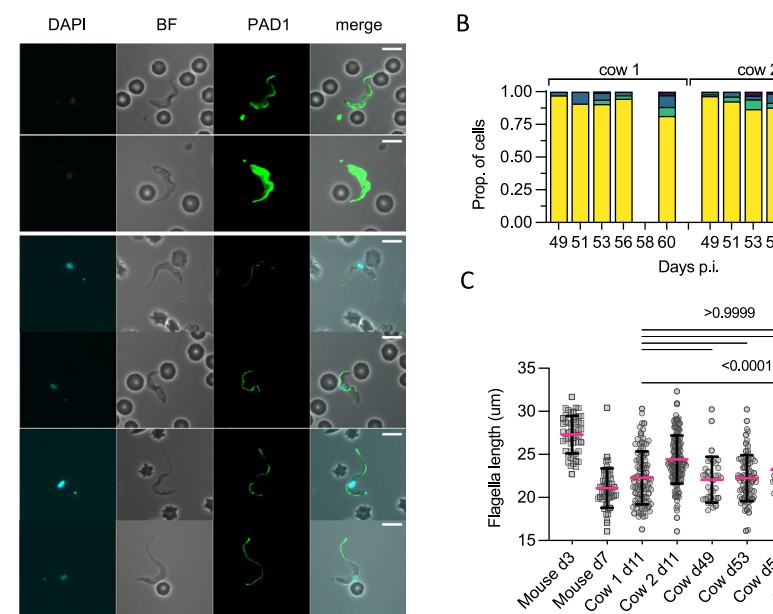

**Fig. 5 | Chronic stage parasites have a morphology consistent with differentiation. A** Representative cell images of parasites isolated from d49 to d60 p.i. Only two parasites (both shown) were scored as PAD1 positive, with most cells showing similar slender-intermediate morphology of those seen on d11 p.i. **B** Kinetoplast (K) and nucleus (N) configuration counts for each sample. **C** The distribution of flagella measurements in both cows in the chronic phase vs controls from d11 p.i. and mouse slender forms (day 3, $n = 1$ mouse) or stumpy forms (day 7, $n = 3$ mice). $P$-values indicated show the result of two-sided Dunn's multiple comparisons tests between cow d11 p.i. (which contains a high proportion of stumpy-like forms) and the samples taken in the late stage of infection (combined cow 1 and cow 2 replicates). Data are presented as mean +/− SD. Source data are provided as a Source data file.

morphologically stumpy forms in the chronic phase parasite population (Fig. 5A).

To examine their developmental status further, the flagella length of chronic phase parasites was compared to that of cow 1 day 11, when a high proportion of stumpy-like parasites was evident by single cell RNA analysis (Fig. 5C). No significant difference in flagella length was found between these time points, with the exception of day 60 when the mean flagella length was longer (mean for both cows = 25.33 μm). Consistently, this time point was characterised by having a higher proportion of 2K2N cells in both cows. Taken as a group, however, the flagellar length of chronic parasites was most similar to mixed populations of parasites containing both actively dividing and non-dividing differentiating parasites, similar to the cluster 1 stumpy-like population.

Thus, the chronic phase of infections contained large proportions of parasites with the shorter flagella associated with stumpy-like forms albeit without PAD1 protein expression, with a minority of actively dividing forms and scarce PAD1-positive morphologically stumpy forms.

## Host specific difference in *T. brucei* transcriptomes

To explore host-specific adaptations of the parasite populations, we compared the transcriptomes of parasites in each of the cow samples to our previous single-cell transcriptomic data generated from either in vitro culture-derived[21] or mouse-derived parasites[13] by pseudo-bulking, whereby transcript counts for each gene were summed across every cell in the population within that sample. The 'In vitro' samples were mixed populations of slender and stumpy forms generated by exposure to brain heart infusion broth (BHI) to trigger stumpy development[21]. Mouse derived samples contained a mix of slender, differentiating and, most prominently, stumpy form parasites isolated on days 7 and 23 of infection[13].

For initial principal component analysis (Fig. 6A), variant surface glycoproteins (VSGs) were removed from the genes used to calculate principal components to avoid grouping by VSG expression, which is

expected to vary across infection. Differential expression tests were then performed with DESeq2 between the different host environments (Fig. 6B–D). All samples were used for each condition (in vitro, mouse or cow) in these comparisons irrespective of time point or infection phase.

From the PCA analysis, it was apparent that *T. brucei* from mouse and culture systems were very similar to each other, although 55 genes had higher expression and 32 showed lower expression in mouse samples when compared to parasites exposed to the BHI quorum-sensing signal in vitro (Fig. 6B, Supplementary data 2). Of these, nine differentially expressed genes encoded different expression site associate genes (ESAGs) and VSGs, perhaps reflecting use of an alternative expression site. ESAG10, which encodes folate transporters, and ESAG6, transferrin receptor subunit, were overexpressed in mouse derived samples, whereas ESAG1 was downregulated compared to BHI generated samples.

In contrast, the transcriptome of cattle derived parasites was clearly distinguished from those of mice and culture (Fig. 6A). Most variance was explained by PC1 (61%), which separated cow samples from mouse or culture samples. Additionally, there was variation between samples taken over the course of cattle infection, and between individual cows at the same time point. Strikingly, after 23 days of mouse infection the transcriptome of cells remained closely related to cultured parasites, even when the proportions of slender and stumpy forms varied between samples. In contrast, parasites from cattle could be discriminated from mice and culture derived parasites based on their transcriptome as soon as 4 days after infection, demonstrating rapid gene expression changes, most likely related to host adaptation, following bovine infection.

Cattle derived samples had far greater numbers of differentially expressed genes in comparison to both in vitro (Fig. 6C) and mouse derived (Fig. 6D) samples. GO term enrichment for genes with higher transcript levels in cattle derived samples (1678 adjusted $p < 0.05$ and fold change >2), showed enrichment for transcripts associated with components of the cell surface, cytosolic ribosomes, chromatin, the

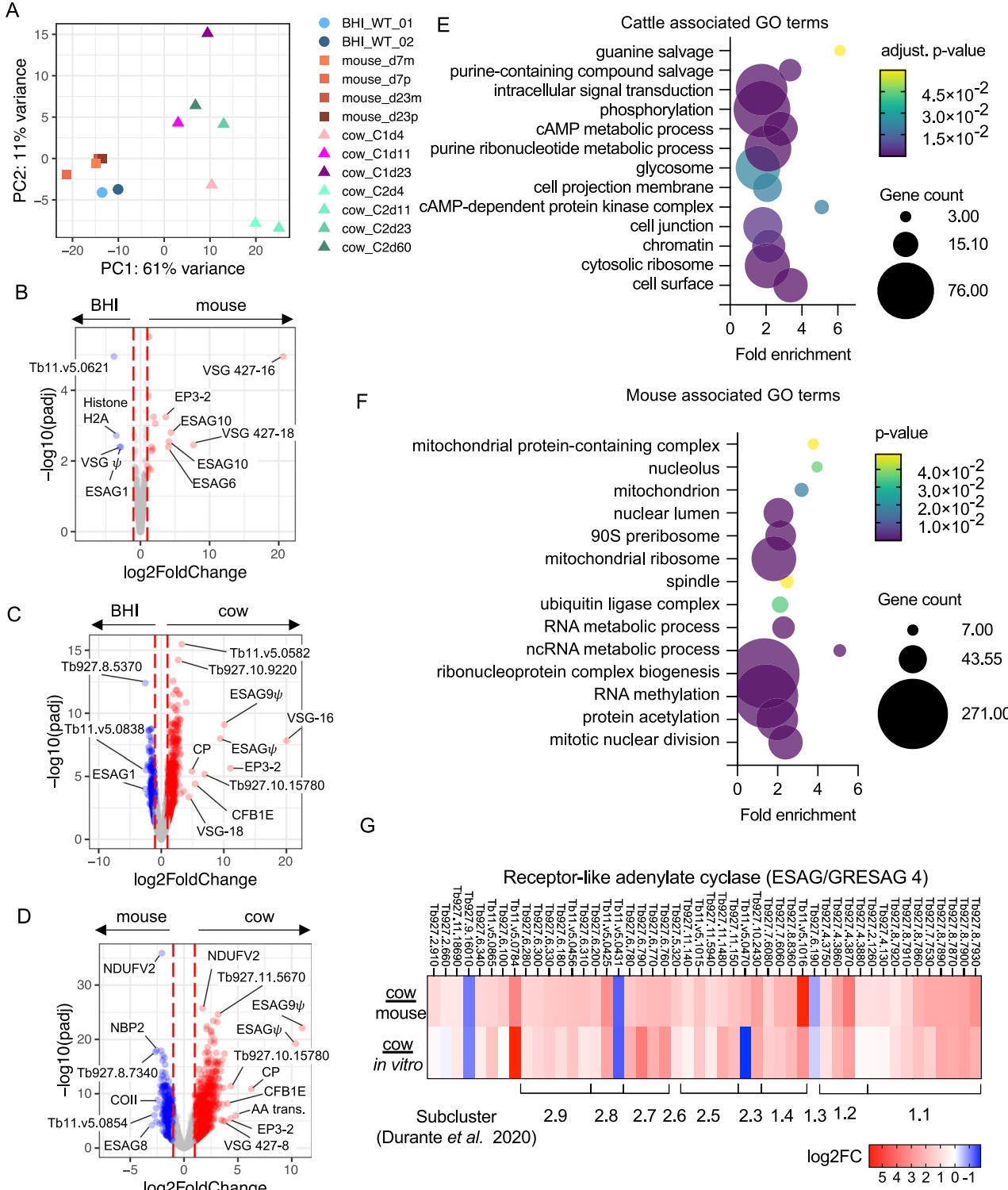

**Fig. 6 | Host specific changes in *T. brucei* bloodstream forms. A** PCA plot of all in vitro (circles, ref. 21), mouse (squares, ref. 13) and cattle (triangles) derived single cell transcriptomic samples, that have been pseudo bulked. Volcano plots of adjusted *p* value (y-axis, −log10(padj) and fold change (x-axis, log2(fold change)) of differentially expressed genes between pseudo-bulked (**B**) BHI treated in vitro and mouse derived samples, **C** BHI treated in vitro and cattle derived samples and **D** mouse and cattle derived samples. In each case all samples and time points, as shown in (**A**), are used for each host environment and padj values are Benjamini-Hochberg adjusted *p*-values from Wald tests. **E** Enriched GO terms in genes over expressed in cattle derived samples compared to mouse derived samples. **F** Enriched GO terms in genes over expressed in mouse derived samples compared to cattle derived samples. GO term adjusted *p* values are Benjamini-Hochberg adjusted *p*-values from Fisher's exact test. **G** Fold change (log2) in ESAG4/GRESAG4 transcript levels encoding receptor-like adenylate cyclase proteins, for parasites isolated from cattle relative to mice (top) in vitro culture (bottom). Clade and subcluster of each gene previously identified[33] are shown below.

flagellum and the glycosome (Fig. 6E). GO terms associated with genes that showed higher transcript levels in mouse derived sample largely related to ribonucleoprotein complex biogenesis, as well at nuclear division and the mitochondrion (Fig. 6F). Predicted cell surface proteins that varied in their expression between infections of mouse or cattle hosts included expression site associated genes (ESAGs). The ESAG classes that varied most commonly between host species were ESAG10 (3/7 annotated ESAG10 genes with detectable transcripts differentially expressed between mouse and cow), ESAG4 (7/18 genes), ESAG2 (9/25 genes) and ESAG8 (3/11 genes) (Supplementary data 2). The ESAGs with the greatest fold-change in transcript levels were two pseudogenes upregulated in cattle samples, ESAG9 (Tb927.11.110) and a non-classified ESAG (Tb929.11.130), that are located together in a likely VSG expression site (Fig. 6D). The ESAGs with highest expression in mouse samples in comparison to cattle are also located in the same VSG expression site, Tb427.BES112 (Fig. 6D and Supplementary data 2). Strikingly, GRESAG4 (gene related to ESAG4), which are located in the core chromosomes and not the VSG expression sites, were also upregulated in cattle derived parasites (Fig. 6G). The ESAG4 and GRESAG4 gene family encodes receptor-like adenylate cyclase proteins (ACs), which can be phylogenetically clustered into two clades and further subgroups[33]. Clades 1 and 2 contain ACs found to be translationally upregulated in bloodstream and procyclic forms, respectively, by ribosomal profiling[34]. Transcripts encoding ACs of both clades were upregulated in cattle derived trypanosomes (31.8% clade 1 and 33.7% clade 2) relative to samples from mouse infections and in vitro culture. Notably, transcripts encoding ACs termed ACP1-6 that are known to localise to the flagellum[35] and in some cases have been linked to social motility[36] were not differentially expressed between host environments. Thus, the observed changes in AC transcript expression levels here do not appear to be linked to life cycle form development. An alternative possibility is that AC upregulation is linked to *T. brucei* adaptation or response to the host immune response. In early infection in mice, the activity of AC enzymes released by lysed parasites inhibits the production of trypanosome-suppressing TNF-α in liver myeloid cells, allowing *T. brucei* to control host early innate immune[37].

## Host specific differences between developmental forms

Given the clear difference between hosts, we next sought to understand developmental form-specific differences between host environments. Attempts to fully integrate data from in vitro, mouse and cattle experiments were inconsistent across integration methods. In the absence of "ground-truth" for these methods we instead used a label transfer approach to predict the cell types in the cattle data (query dataset) using either in vitro (Fig. 7A, ref. 21) or mouse (Fig. 7B, ref. 13) parasite cluster labels from our previous analyses as the reference datasets. Providing evidence for the validity of using label transfer, a comparison of monomorphic slender form transcriptomes[22] to in vitro derived slender and stumpy forms[21] demonstrated that 97.9% of monomorphic slender forms labelled as slender, as expected (Supplementary Fig. 7).

Consistent with our earlier analysis (Fig. 1D), predicting the cell types using the in vitro observed developmental forms as the reference (Fig. 7A) resulted in the labelling of cattle cluster 1 as stumpy forms and clusters 2 and 3 as slender forms (Fig. 7C). Cluster 0 was labelled as a mix of slender and stumpy forms when compared to the in vitro reference, likely due to the low expression of cell cycle marker genes (Fig. 1F). Previous analysis of *T. brucei* isolated from the mouse bloodstream on days 7 and 23 of infection[13] revealed that nearly all parasites were arrested stumpy-like forms at these time points, with only 2.1% of the captured parasites being slender forms (cluster M3, Fig. 7B). In that study a clear stumpy population (cluster M0) was identified as the most common form in the mouse bloodstream (82.8%), and two other clusters (clusters M1 and M2) were identified that may represent "intermediate" forms transitioning between

slender and stumpy extremes[13]. Using the label transfer method to predict the corresponding cell types between cattle and mouse derived parasite samples, we identified 70.9% of cattle cluster 1 (stumpy-like) as most similar to mouse cluster M0 stumpy forms (Fig. 7D). Smaller proportions of cattle cluster 1 were assigned as cell types M1 (7.3%) or M2 (5.3%), suggesting these putatively intermediate parasite forms were present but in low proportions in cows at the points of infection analysed. Therefore, the stumpy-like forms found in the cattle bloodstream were most similar in terms of their transcriptome to mature stumpy forms found in mouse infections, despite the differences in morphology and lack of PAD1 expression. Matching the comparison to the in vitro data, label transfer for cow cluster 0 resulted in a mix of slender (M3) and stumpy (M0) labels. Additionally, a proportion of all cattle clusters preferentially labelled as mouse cluster M4 (Fig. 7D). Mouse population M4 was observed in lower levels in previous murine experiments (Fig. 7B) and biological processes linked to this distinct subpopulation were unclear[13].

The transcript level fold change between stumpy/stumpy-like (positive fold change (FC) values) and slender/slender-like (negative FC values) populations was next quantitated for each dataset individually (Fig. 7E–G and Supplementary data 3). Genes were filtered for those significantly variable between forms (adjusted $p < 0.05$ and log fold change > 0.5) in at least one dataset and then the changes were compared across environments. The highest correlation was evident between in vitro and mouse-derived datasets ($r = 0.73$, $p < 0.05$), with changes between slender and stumpy-like forms showing slightly lower correlation between these data and cattle samples (cow vs. mouse: $r = 0.59$, $p < 0.04$; cow vs. in vitro: $r = 0.68$, $p < 0.05$).

Comparing slender-associated transcripts, 246 genes were upregulated in slender forms across all datasets (Supplementary data 3). The top common markers consisted of genes linked to cell proliferation (e.g. histone H2B, chromosome passenger complex 1 and cytoskeleton components), numerous RNA-binding proteins (e.g. RBP10, ZC3H38, ZC3H39, ZC3H31) and a non-coding RNA Tb1.NT.27. Just 13 genes were uniquely upregulated in slender-like forms in the cattle data only compared to both in vitro and mouse data (Fig. 7E, F). Of these, five are also markers of cow cluster 3 described above: a 2-oxoglutarate dehydrogenase E1 component (Tb927.11.9980), aquaglyceroporin 1 (Tb927.6.1520), a calpain-like cysteine peptidase (Tv11.v5.1011) and two hypothetical proteins (Tb927.11.18710 and Tb927.7.790). The remaining eight genes encode five hypothetical proteins (see Supplementary data 3), a second putative calpain-cysteine peptidase (Tb927.11.1130), F-box protein FBP75 (Tb927.5.700) and a putative calcium motive p-type ATPase (Tb927.9.15460).

A common stumpy-associated transcriptome was also clear (Fig. 7E–G), with the most conserved stumpy/stumpy-like markers including mitochondrial-encoded cytochrome oxidase subunit 2 (COII), procyclin proteins (EP1, EP2, GPEET), PAD2, multiple copies of retrotransposon hotspot protein 4 (RHS4) and several hypothetical proteins (see, Supplementary data 3). 422 genes showed higher transcript levels in stumpy-like forms in cattle, but not in stumpy forms isolated from mice or in vitro. These included cAMP response protein 9 (CARP9; Tb927.8.4640), amino acid transporter 1 (AATP1; Tb927.8.7640), ESP9 (enriched in surface-labelled proteome protein 9; Tb927.9.11480), two putative metallopeptidase (FtsH33, Tb927.4.3300 and Tb927.11.3490), seven adenylyl cyclase (GRESAG 4.4) copies and members of the PAD array.

Given these host-specific differences, we attempted to adapt cultured *T. brucei* strains to cattle serum in vitro and assess morphological parameters using microscopy. Parasites could not be adapted to 20% cow serum as a replacement for 20% foetal calf serum (FCS) in HMI-9 medium. However, Antat 1.1 90:13 *T. brucei* could be successfully adapted to a mixture of 10% cow serum and 10% FCS in HMI-9 after one week of culture, at which point the parasites exhibited the same growth rate as those cultured in 20% FCS (Supplementary Fig. 8A). At

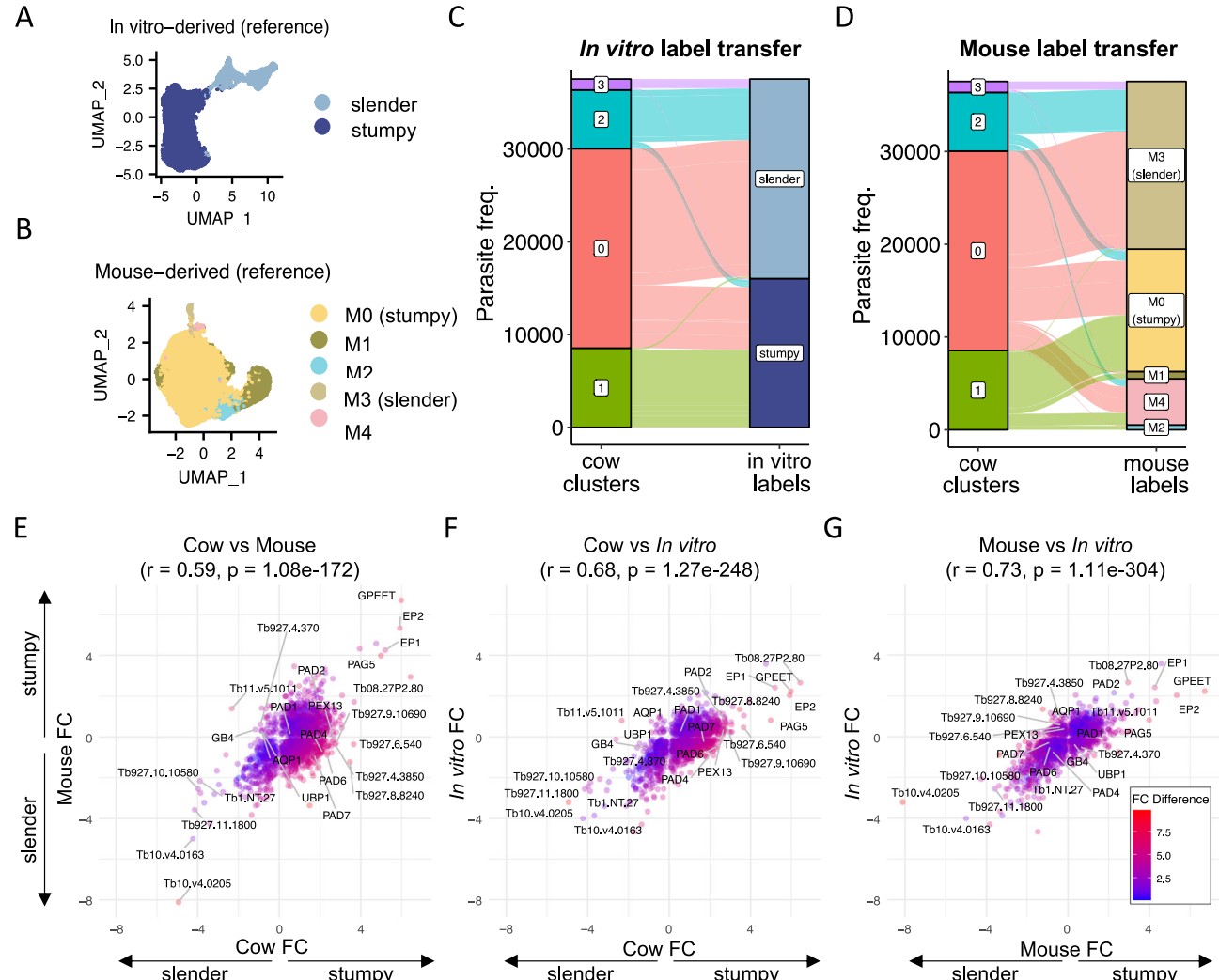

**Fig. 7 | Comparison of *T. brucei* slender and stumpy-like forms isolated from mouse and cattle bloodstreams. A** UMAP of previously described scRNA-seq analysis of *T. brucei* forms exposed to BHI to induce stumpy development in vitro[21]. **B** UMAP of previously described scRNA-seq analysis of *T. brucei* isolated from the mouse bloodstream used as a reference for label transfer[13]. Slender (M3, light brown) and stumpy (M0, yellow) clusters are highlighted. **C** Proportions of each cattle-derived cluster (left) and the corresponding predicted cell type using in vitro reference (right, as shown in **A**). **D** Proportions of each cattle-derived cluster (left) and the corresponding predicted cell type using the mouse reference (right, as shown in **B**). Fold change in transcript levels between stumpy- and slender-like cell types for cow *vs*. mouse (**E**) cow *vs*. in vitro (**F**) and mouse vs in vitro (**G**). In each case, results from the Pearson correlation test (R), along with the corresponding two-sided *t*-test *p*-value, are shown in the title. Points are coloured by the difference in fold change between the two compared datasets.

high density (>1.5 × 10^6 parasites/ml), there were no observable differences in the proportions of 1N1K parasites or in flagellum length (Supplementary Fig. 8B, C). PAD1 staining revealed no positive parasites in 20% FCS, and 7 out of 245 parasites were PAD1-positive in the 10% FCS/10% cow serum condition across three replicates. Therefore, the addition of uninfected cattle serum alone to in vitro cultures of *T. brucei* does not recapitulate the characteristics observed in parasites isolated from cattle.

## Discussion

Almost all of what is known about parasite development in mammalian trypanosome infections has come from in vitro culture and rodent models. Very little is known about how host and parasite interact to impact development in more clinically relevant systems. Here, we uncovered the presence of non-dividing stumpy-like forms that vary as a proportion of the cattle bloodstream population during infection and persist into chronic infection at a significant level. Unlike rodent infections and in vitro models, the appearance of these forms was not linked to the highest parasitaemia levels, classical stumpy morphology

or expression of the PAD1 marker protein. Yet, at almost all the timepoints we monitored, few parasites were visibly proliferative, and many had flagellar lengths consistent with the morphology of intermediate/differentiating forms. Thus, during a *T. brucei* infection the cattle bloodstream contains replication competent slender forms as well as a significant population of non-replicative parasites with a stumpy-like transcriptome and shortened flagella. Whether these latter forms are (a) an intermediate stage committed to differentiation to classical stumpy forms, (b) a reversibly transient stumpy-like state that can revert to replicating slender forms, or (c) a fully and terminally differentiated form that is distinct from morphological stumpy forms in mice, remains to be investigated. Nonetheless, these observations reveal several features of cattle infection dynamics, and pose new questions: firstly, why do parasites with a 1K1N configuration predominate at low parasitaemia despite the abundance of apparently replication-competent slender-like forms; secondly, if differentiation begins for a subpopulation of blood parasites, why are fully transitioned morphological stumpy forms rare; and, thirdly, do the stumpy-like forms we observe contribute to transmission?

The classic view of trypanosome infection dynamics entails undulating waves of infection where rapidly dividing slender forms expressing a dominant antigen type prevail, until either a density-dependent quorum-sensing signal triggers differentiation into arrested stumpy forms or host immunity eliminates the bulk of the population, after which new antigen types repopulate the infection. This accurately reflects the first waves of parasitaemia, particularly in mouse infections, although a more complex view of VSG dynamics is now established based on higher resolution analyses of VSG expression profiles[38]. However, at parasitaemias below a density required to activate quorum sensing, only immune-mediated clearance of slender forms would be expected to operate in controlling parasitaemia. With the infection dynamics observed in cattle infections, according to parameters derived from rodent or in vitro systems, all parasites present should be replicative slender forms, which would maintain the infection through multiplication and the generation of antigenic diversity. However, our combined data suggest that the bloodstream population in cattle is heterogeneous despite the low parasitaemias; we find cell types that are configured in 1K1N (up to 97% in the chronic infection phase), have a shorter flagellum than would be expected for slender forms in the logarithmic growth phase[32], and express stumpy-associated transcripts. This invokes a requirement for the parasites to undergo developmental adaption in the mammalian host, even at low overall cell density in blood.

A number of scenarios could account for transmission-adaptation and arrest occurring in cattle, despite parasitaemia below the in vitro threshold for density-dependent differentiation ($-1 \times 10^6$/ml). Firstly, it is possible that quorum-sensing pathways are triggered at lower density in cattle. In mice, *T. brucei* slender forms release oligopeptidase B and metallocarboxypeptidase 1 (among other peptidases) into the environment, generating the oligopeptide signal for stumpy formation[39,40]. Although the level of transcripts for these peptidases was not seen to be elevated in our cattle samples, these enzymes or their oligopeptide products may be more active, or more stable in cattle, allowing quorum sensing at lower parasitaemia. Secondly, stumpy-like form development may be triggered by alternative signals that are not only dependent upon parasite density. For example, an inflammatory environment may generate an oligopeptide signal driving development[41], or an environmental metabolite may promote stumpy formation in a quorum sensing-independent mechanism. Thirdly, parasites may achieve much higher density outside the blood in cattle, with stumpy-like parasites then seeding into the circulation. This latter scenario in particular has some experimental support, as analysis of mouse infections has previously demonstrated that there are apparently insufficient proliferative bloodstream forms to sustain the parasitaemia, with proliferation and development predicted to be occurring in the tissues[13]. This is further supported by the appearance of novel VSG types in blood after their emergence in the extravascular spaces[19]. The tissue-distribution of *T. brucei* in cattle is poorly understood[42] but recent work from cows in Ghanaian abattoirs indicates that parasites are more often detected in tissues than in blood alone[16]. It is also noteworthy that samples analysed here were taken from the major venous circulation. The nature of *T. brucei* found in the small vessels of the microcirculation, that within the skin will be accessible to tsetse flies, may again be different and requires investigation.

The absence of fully morphologically stumpy forms in the cattle blood may be contributed to by an inability of morphological stumpy forms to successfully enter the circulatory system from tissue locations, or incomplete or delayed progression through the hierarchy of events that generate classic stumpy forms in cattle. In earlier analyses, we mapped the events of stumpy formation in mice, revealing that a stumpy transcriptome precedes cell cycle arrest, PAD1 expression and morphological development to stumpy forms[13]. Similarly, in cattle, parasites may progress along the pathway to stumpy formation but not fully develop to its latter stages, potentially predominating as uncommitted forms with the developmental flexibility to return to proliferation as slender forms or commit to arrest as stumpy-like forms. Entry into the fly may then activate PAD protein expression, which is thermoregulated in the case of PAD2[26], to initiate development to procyclic forms from cluster 1 parasites with elevated PAD mRNA. Unfortunately, the inability to culture parasites ex vivo after cattle infection has limited the ability to explore whether parasites are committed to arrest or can return to proliferation; however, the molecular basis of this adaptation will be an interesting area of exploration. While bulk transcriptomics has previously revealed a developmental pattern of PAD array transcript expression from in vitro generated slender and stumpy forms through the tsetse fly stages[43], the broader range of PAD upregulation observed in single-cell transcriptomics of mixed bloodstream form populations from cattle also remains to be fully understood. An alternative explanation for the lack of morphologically stumpy forms in cows may be differences in parasite clearance rates between hosts. Immune exhaustion caused by loss of B cells has been demonstrated in chronically infected mice, leading to reduced ability to remove parasites and their eventual death[44], this occurring at a time at which mice appear to have reduced ability to clear specific antigenic variants[38]. If parasites can survive longer before their removal in mice than cows, then a full morphological transition, which may take days to complete, would be more likely to be observed in mice.

Our results highlight differences between mouse and cattle infections, and between early and chronic phases. In the early stage of infection, mouse and cattle infections both show relatively high initial parasitaemia, which is controlled by the host. However, subsequently, cattle infections differ in their much lower parasitaemia and show less evidence of clearly discriminated slender and stumpy morphotypes. These distinct acute and chronic phases may serve to promote establishment through the rapid proliferation of slender forms early on, where higher antigenic diversity serves to overcome any pre-existing field immunity and parasite growth can be restricted by quorum sensing in the first peak. Subsequently, the chronic phase would be comprised of parasites that can either sustain infection through proliferation and antigenic diversity or be adapted for transmission[45]. These "pre-committed" cell types that predominate in chronic infections might reflect a bet-hedging strategy for the parasites, optimised for survival or transmission where tsetse uptake is uncertain and long-term infection maintenance is required.

The distinctions between parasite transcriptomes in mice and cows were also evident as early as day 4 post infection, highlighting rapid gene expression changes, most likely related to adaptation to the host environment. This was also supported by our inability to culture parasites ex vivo from cows beyond this time point, despite clear evidence that large proportions of these parasites had a slender-associated transcriptome consistent with replicative capacity. Notably, addition of adult cow serum to cultured parasites did not mirror this rapid in vivo adaptation, indicating more complex host-parasite interactions. In contrast, the gene expression profiles of culture and mouse derived cells were much more similar, and it is notable that mouse-derived ex vivo cells will readily grow in the conditions that were unsuccessful for cattle derived parasites. The few genes with differing transcript levels in cattle slender forms compared to mouse and culture derived slender forms were largely linked to glycolysis, perhaps suggesting a switch in energy metabolism dynamics in response to the highly diverging host environments. However, such rapid adaptions will likely also involve protein level changes that could not be investigated in this study.

We identified a subpopulation of slender forms across the cattle samples with upregulation of metabolic genes, including succinyl-CoA synthetase, glycerol kinase, oxoglutarate dehydrogenase, and pyruvate dehydrogenase. Mitochondrial ATP production via substrate-level phosphorylation by succinyl-CoA synthetase, shown to be essential in vivo[46], may be fuelled by succinyl-CoA derived from

pyruvate dehydrogenase (via glucose) or oxoglutarate dehydrogenase (via 2-oxoglutarate from glutamine[47,48]). However, transcripts for glutamine synthetase and glutamate dehydrogenase were not enriched in cluster 3 (Fig. 2), suggesting elevated mitochondrial ATP production. The upregulation of glycerol kinase may instead support gluconeogenesis, as shown in bloodstream[49] and procyclic[50] forms, supported by a modest (non-significant) increase in fructose-1,6-bisphosphatase (enzyme 5, Fig. 2C). Given the lower glucose levels in ruminants (-2 mM vs. -5 mM in non-ruminants), glycerol may serve as an alternative carbon source during cattle infections.

In summary, we have for the first time characterised the parasite populations that comprise acute and chronic phases of trypanosome infections in cattle and related these to established paradigms in mice and in vitro. We observed that infections are sustained at low parasitaemia and comprised of parasites that are not morphologically stumpy but express transcripts indicative of that developmental form. By establishing the molecular characteristics of parasites from the chronic phase of the infection, we have also extended beyond previous rodent-based laboratory studies of acute infections and broadened analysis to parasites in chronic infections in the natural host, representative of most infections in the field.

## Methods

### Cattle infections, parasitaemia and microscopy

Two male post-weaning Holstein Friesian calves aged 4–6 months were housed in vector proof containment (BSL2) conditions at the Large Animal Research and Imaging Facility of the Roslin Institute, University of Edinburgh. To initiate calf infections, blood containing *T. b. brucei* Antat 1.1 90:13 was freshly harvested from an infected donor mouse and quantified. Blood containing $1 \times 10^6$ parasites in 1 ml from this same stock was used to infect each calf via the intravenous route. As blood parasitaemia was frequently extremely low and below the limit of accurate quantification using buffy coat preparation, we used a conversion between buffy coat and haemocytometer counts (parasites/ml = 29,478 × parasites/field of view [FOV]) on all data points to infer parasitaemia at time points where parasites were detected but in extremely low numbers (<1 parasite per buffy coat FOV; Supplementary Fig. 1). Parasite cell line provided by K. Matthews.

### Parasite purification and scRNA-seq

200 ml of blood was collected for each scRNA-seq sample on days 4, 11 and 23. To obtain a pure parasite sample, a combination of red blood cell (RBC) lysis and DE52 column purification was used. Each whole blood sample was aliquoted into eight 50 ml falcon tubes (-16 ml per falcon) and centrifuged at $1200 \times g$ for 15 min without breaks to separate blood. The plasma was carefully removed, leaving RBCs and buffy coat behind where parasites will be located. A hypotonic lysis buffer (0.3% NaCl in $H_2O$) was added to each tube to a total volume of 45 ml and inverted. 5 ml of 10X PBS (Phosphate Buffered Saline) was then added and tubes inverted to halt RBC lysis. Samples were centrifuged at $80 \times g$ for 20 min to pellet trypanosomes and retain as much of the blood cells in the supernatant as possible. Supernatant was then removed, and each pellet was combined by resuspending all in 10 ml of PSG (1X PBS + 1% D-glucose) and combining into one 50 ml tube. The combined sample was centrifuged at $400 \times g$ for 10 min and supernatant removed, leaving 2–3 ml of volume to resuspend the pellet in. The resuspension was mixed 1:1 with DE52 cellulose slurry and then applied to a glass filter column containing -100 ml of DE52 cellulose pre-washed with warm PSG. Columns were slowly washed with -100 ml of PSG to obtain pure trypanosomes and retain blood cells in the columns. Eluted parasites were then centrifuged at $400 \times g$ for 10 min to pellet. Supernatant was removed to leave -1 ml of PSG to resuspend the pellet. Parasites were then counted with a haemocytometer and adjusted to 1000 cell/μl by transfer to a clean Eppendorf, further centrifugation ($400 \times g$ for 10 min) and resuspension in an

appropriate volume of PSG. Cells were counted again and adjusted as needed before 20,000 were loaded onto the Chromium (10X Genomics) controller for scRNA-seq. The Chromium Next GEM Single Cell 3′ Gene expression kit v3 was used and sequencing was performed with Illumina NextSeq 2000 to generate 28 bp × 130 bp paired reads to a depth of at least 63,415 mean reads per cell. In the case of cow 2 day 23, two libraries were prepared and sequenced as low droplet formation was apparent in this sample.

### scRNA-seq processing and cluster analysis

scRNA-seq data were mapped to the *T. brucei* TREU927 reference genome with extended UTRs to ensure all reads mapped were correctly assigned to a gene coding region. The kDNA maxicircle and *T. brucei* Lister 427 telomeric expression site contigs were also included to create a "hybrid" reference, as these regions are not present in the TREU927 ref. [51]. Mapping was performed with Cellranger count and default settings to generate the genes counts matrices. Each sample was inspected and filtered to remove outliers that are likely poor-quality transcriptomes or multiplets (Supplementary Fig. 2). Outliers with high kDNA (mitochondrial) and ribosomal RNA transcripts were also removed as these both indicate poor quality transcriptomes.

Each sample was normalised, and log transformed with Scran[52] independently. Variable genes were identified by both Scran and Seurat[53] methods and the common genes (with VSGs removed) were assigned as variable genes for each sample. Cattle samples were integrated using Seurat, scaled and PC analysis performed. UMAP was performed for visualisation using the top 20 PCs, as was nearest neighbours and clustering analysis.

For slender, stumpy and cell cycle phase scoring, average expression score of previously identified marker genes lists[21,22] was calculated using the Seurat function *AddModuleScore*. This function calculates the average expression of all markers per cell, compared to a set of control genes. A score >0 indicates higher expression for the marker gene set compared to the control genes. For assigning the most likely cell cycle phase with these markers, if all phase scores failed to reach the threshold of 0, the transcriptome was "Unlabelled", otherwise the top scoring phase was assigned.

Cluster marker genes were identified using MAST[54] differential expression tests to find genes overexpressed in one cluster compared to all other clusters. GO term analysis was performed via the TriTrypDB[55] website.

### Cross scRNA-seq dataset comparisons

The scArches package[56] was used to identify the cell types in cattle derived transcriptomes based on in vitro and mouse derived slender and stumpy *T. brucei* cell types that have been characterised previously. Independently, reference datasets were normalised and logp1 transformed with Scanpy[57]. 1500 variable genes were selected with Scanpy and the data subsetted to only use counts from these genes. A model was then created with scPoli[58] and trained on the reference data set (in vitro or mouse derived) using previously defined clusters as the cell type key. Models were then used to predict cell types in the cattle query data. To calculate fold change in transcript levels for all genes between slender and stumpy/stumpy-like clusters, FindMarkers() from Seurat was used with all filters set to 0, and the test.use set to "MAST". Top markers for each cluster were defined as having an adjusted $P$ value < 0.05, an average log2(fold-change) > 0.5 and being detected in at least 5% of the cells in the cluster under investigation.

### Pseudobulk RNA-seq analysis

To directly compare cattle data to previous scRNA-seq datasets of in vitro and mouse derived slender and stumpy forms, these data were remapped to the "hybrid" reference genome and pre-processed in the same steps used in the relevant previous publications (where the telomeric contigs were not previously included for mapping). For

consistency, all QC filtering was performed exactly as previously described for each. For each sample, total counts for each gene across all cells in the sample were summed to generate one total transcript count per gene. VSGs and contaminating ribosomal RNA transcripts were removed before DESeq2[59] was used for PC analysis using the top 2000 genes. Individual DESeq2 differential expression tests were performed to compare conditions (i.e. host environment).

### Ex vivo replating assay

The plating assay described previously[13] was used for assessment of ex vivo proliferation of parasites following blood sampling and parasitaemia scoring on days 4, 11 and 23 post infection. A Neubauer hemacytometer was used to accurately count the parasites, and a final stock sample with a fixed concentration of 1 cell/μL in 5 mL total volume was made. The stock was then used to seed round-bottom 96-well plates with the following cell numbers: 50/well, 20/well, 5/well, 2/well, and 1/well. The wells were filled with HMI-9 media to 100 μL total volume. After seeding, the plates were incubated for 5 to 7 days before each well was assessed for an outgrowing culture of parasites in a binary fashion. Twelve replicates were made of each seeding. The number of wells that developed outgrowth was counted and the percentage growing for each condition was calculated. Since only one competent cell is required for outgrowth, the probability (P) theory was used to calculate the estimated proportion of replication-competent cells using the following formula: P(at least one cell replicative) $= 1 − P$(failure in one trial)$^n$, where $n$ is the total number of trials. An average of the values of replication-competent cells across the seedings was used where there was variation in outgrowth across the dilutions, excluding conditions where either all or none of the wells had outgrowth as these do not provide the required variation.

### PAD1 immunofluorescence microscopy

For scRNA-seq time points (4, 11, 23 and 60 days p.i.), purified parasites not required for scRNA-seq or ex vivo replating were used for microscopy. For chronic infection stages (49–58 days p.i.) parasites were purified from 5 ml blood samples using DE52 cellulose slurry columns as described above. Parasites resuspended in PSG were airdried onto glass slides before fixation and storage in methanol at −20 °C. To compare anti-PAD1 staining in pleomorphic and monomorphic *T. brucei* strains, Antat 1.1 90:13 and Lister 427 parasite lines were culture in HMI-9 with 20% FCS, either with or without 15% brain heart infusion broth (BHI). At 24 and 48 h parasites were washed in PSG buffer and airdried to glass slides before methanol fixation at −20 °C. In all cases slides were washed in PBS after methanol fixation, before blocking with 2% bovine serum albium (BSA) in PBS for 1 h. Slides were stained with anti-PAD1 antibody (1:1000 in 0.2% BSA in PBS) for 1 h, washed three times in PBS, and then stained with secondary antibody Alexa Fluor™ 488 goat anti-rabbit (1:1000 in 0.2% BSA) for 1 h. Finally, slides were stained with DAPI (100 ng/mL) for 5 min before washing three times in PBS, before mounting with Fluoromount G (Thermo Scientific) and sealing.

Images were taken using a Zeiss Axioscop2 microscope at 63x magnification. PAD1-positive cells were classed based on any PAD1 staining that was not restricted to only the flagella or flagellar pocket. The length of the flagella was measured from all images of cells in 1K1N using ImageJ[60]. The free hand tool was used to trace the flagella from the flagellar pocket to its termination point using a combination of phase contrast and PAD1 staining (which consistently stains the flagella in all cell types both stumpy, slender and intermediates). The length in pixels was converted to μM in ImageJ using the conversion one pixel = 0.102 μM.

### Treatment of in vitro *T. brucei* cultures with cow serum

Fresh, non-heat treated, non-filtered serum from an adult Holstein Friesian cow was added to HMI-9 media at either 20%, or 10% in combination with 10% FCS. Supplemented medias were used to generate culture at $1 \times 10^5$ parasite/ml. Cultures were incubated at 37 °C and monitored daily, diluting to ensure density remained below the quorum-sensing threshold when parasites arrest in culture. Only culture in combined 10% FCS and 10% cow serum allowed proliferation in vitro. After one week, these adapted parasites were diluted to $1 \times 10^5$ parasite/ml alongside parasites maintained in 20% FCS and growth was compared over 48 h. After 48 h parasites were prepared for microscopy analysis as described above.

### Ethics statement

Animal experimental infections were carried out in the Large Animal Research and Imaging Facility at the Roslin Institute, University of Edinburgh, under United Kingdom Home Office Project License number PE854F3FC. Cattle sampling for obtaining serum used for in vitro culture experiments was undertaken under Home Office Project License number PP0612543. Studies were approved by the Roslin Institute (University of Edinburgh) Animal Welfare and Ethical Review Board (study number L553). Care and maintenance of animals complied with University regulations and the Animals (Scientific Procedures) Act (1986; revised 2013).

### Reporting summary

Further information on research design is available in the Nature Portfolio Reporting Summary linked to this article.

## Data availability

The scRNA-seq data generated in this study have been deposited in the European Nucleotide Archive database under accession code PRJEB66078. The processed scRNA-seq data are available at zenodo.org under https://doi.org/10.5281/zenodo.14515536. All other data generated in this study are provided in the Supplementary Information/Source data file. The in vitro and mouse derived scRNA-seq data used in this study are available in the European Nucleotide Archive database under accession codes PRJEB41744 and PRJEB60851, respectively, and as processed data at zenodo.org under https://doi.org/10.5281/zenodo.14515536 Source data are provided with this paper.

## Code availability

All code needed to reproduce this analysis is accessible at zenodo.org under https://doi.org/10.5281/zenodo.14515536.

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

## Acknowledgements

This work was supported by a Wellcome Trust Investigator award to K.M. (221717/Z/20/Z), a Wellcome Trust collaborative award to K.M. and L.M. (206815/Z/17/Z), a Sir Henry Wellcome Fellowship awarded to E.B. (218648/Z/19/Z) and a Roslin Institute Pump Priming BBSRC grant to L.M., P.S., K.M. and E.B. (sub-award from BBS/E/RL/230002C). L.M., E.P., C.V. and P.S. are funded through core support to the Roslin Institute by the United Kingdom Biotechnology and Biological Sciences Research Council (BBS/E/RL/230002C); P.S. is supported by a BBSRC Discovery Fellowship (BB/X009807/1); E.B. is supported by an MRC Career Development Award (MR/Z504786/1). We would like to acknowledge the assistance of staff members at the Large Animal Research and Imaging Facility, University of Edinburgh; specifically, James Nixon, Peter Tennant and Adrian Ritchie, who provided invaluable expertise and assistance in the experimental cattle infections. We would also like to thank Stefano Guido of the University of Edinburgh for expert advice, and Richard McCulloch and his group at the University of Glasgow for accommodating the single-cell experiments. Finally, we would like to thank Julie Galbraith of Glasgow Polyomics for facilitating this unpredictable study and providing expert scRNA-seq library preparations and sequencing.

## Author contributions

P.S., E.B. and S.L. developed the approach for parasite isolation. E.P., C.V., E.B. and S.L. produced the data in this study. S.L. performed microscopy analysis; E.B. performed scRNA-seq analysis. S.L. and E.B. produced the manuscript, along with K.M. and L.M. and comments from co-authors. All authors contributed to the interpretation of results, discussion and final manuscript.

## Competing interests

The authors declare no competing interests.
