## [Transparent Peer Review file · Nature Communications]

***Trypanosoma brucei* cattle infections contain cryptic transmission-adapted bloodstream forms at low parasitaemia**

Corresponding Author: Dr Emma Briggs

Version 0:

Reviewer comments:

Reviewer #1

(Remarks to the Author)

In this nice manuscript, the authors have characterized the parasite populations that comprise acute and chronic phases of trypanosome infections in cattle and related these to established paradigms in mice and in vitro. They observed that infections are sustained at low parasitaemia and comprised of parasites that are not fully morphologically stumpy but express transcripts indicative of that developmental form. By establishing the molecular characteristics of parasites from the chronic phase of the infection, they have smartly extended beyond their previous rodent-based laboratory studies of acute infections and broadened analysis to parasites in chronic infections in the natural host. The discussion is very well written, putting the emphasis on the hypothesis drawn by this in-depth descriptive work. I only have very few comments.

Main comment

In the title as well as along the entire manuscript, the authors assume that 'transmission-adapted bloodstream forms' are necessarily characterized by the expression of PAD1 over the entire plasma membrane. Is it correct? The authors state that 'No parasites showed expression of the PAD1 surface protein' on D11, however, figure 2 shows a clear PAD1 signal at the flagellum. In Larcombe et al. PNAS 2024, PAD1 seems to first be detected at the flagellum during the course of the differentiation. This is a pattern also frequently seen by IFA on AnTat1.1E BSF from mice (Calvo Alvarez et al. PLoS Path 2023). In total, the so-called early ST-like forms described in the manuscript present all characteristics of ST forms but a PAD1 expression restricted to the flagellum. Couldn't it be sufficient for progressing through the parasite cycle after transmission? In that case, transmissible forms wouldn't be so cryptic.

Minor comments

Introduction

- L63-64: This is correct but it was not demonstrated for the first time in the cited reference.
- The preponderance of extravascular parasites during an infection in the mammalian host could be already explained in the introduction as it is a major biological trait of Tbb biology.

Results

- Figure 1: Could the authors show the PAD1 expression levels in panel E? Panel F apparently shows subtle differences that are difficult to detect.
- Were Factor H Receptor transcripts also enriched in Cluster 1?
- Considering their surprising profile (upregulated TCA and pyruvate metabolism transcripts), what could be the nature / function of cells in cluster 3?
- L194: Do you mean 'the proportion of proliferative parasites'?
- As previously described, e.g. in Pyana et al. PLoS NTDs 2011, strain isolation in culture or adaptation in laboratory mice are not trivial. In Van Reet et al. exp parasitol 2011, the authors used methylcellulose and human serum to support the continuous in vitro propagation of the bloodstream form. KIVI could also be useful to simultaneously assess the presence of parasites and their ability to differentiate into PCF in vitro (Truc P et al. Trans R Soc Trop Med Hyg 1992).
- Could PAD4, PAD6 and PAD7 ensure a redundant function in the differentiation process when PAD1 is barely expressed?

Discussion

- The authors fairly mention extravascular parasites in this section, but as all experiments were performed on parasites from the major venous circulation, could parasites in the micro-circulation be also different in terms of proliferation and transmissibility?
- L445: "The absence of fully morphologically stumpy forms in the cattle blood may be contributed to by an inability to successfully enter the circulatory system from the tissues, or by a more limited progression through the hierarchy of events that generate classic stumpy forms." Again, why would it be necessary for parasites to reach the fully morphologically stumpy forms if a ST-like form (PAD1 at the flagellum only) is sufficient for transmission?

NB: Some thoughts... Could the flagellum possibly be the first region of the cell surface where key proteins such as PAD1, and possibly also VSGs (e.g. during metacyclogenesis in Rotureau et al. Dev 2012) are de novo exported?

Reviewer #2

(Remarks to the Author)
Larcombe et al

This is another instalment in the studies by Briggs et al using scRNA-seq to characterise trypanosome differentiation. I think I have read most of these papers. This one looks at cattle infections using two animals and multiple time points, and which is a commendable goal as we have become aware of the limitations of small rodent models and their failure to accurately reflect infections in larger animals.

The present work is technically sound, but my major issue is with presentation and interpretation. For a dataset such as this, providing a narrative is critical, as there are so many ways the data could be processed, and a justification really is needed for each analysis. Essentially, unpack, show some narrative and explain. I have some specific points that may help with moving the study forward. There does not appear to be a question, and there is no mechanistic insight and the abstract needs rewriting. I'd also argue that perhaps presenting the data in a more open minded fashion, with the obvious major thing being that the cow and murine models are hugely different, would make for a better story as well as perhaps a more impactful finding.

L22 Probably need to be more specific on subspecies here.

L34 Obviously a key observation, but how is it that we know parasites are not replicating? Deserves a brief justification.

I find the definition of differentiation extremely vague here - on the one hand we lack morphological or marker protein expression and on the other we have a shortened flagellum. So, does this mean that the shorter flagellum is NOT a differentiation marker?

The idea that the transcriptome reflects the overall state of the cell seems flawed here as the proteome is the real thing. Perhaps the transcriptome per se is not a fair reflection of the cell's status.

We have multiple terms here - stumpy, stumpy-like, extreme stumpy.... None of which have a precise delineation. This leads to some confusion as well as a somewhat uncomfortable sense that maybe even the author's are unclear on this. I appreciate we are not dealing with binary events here and that diversity is a major feature of biology, but the definition of terms is critical.

F1 - Why number 0 to 3 and not 1 to 4? Reason for big difference between cow 1 and 2 on day 11, and to lesser extent elsewhere? Panel D not aligned and based on the text is unclear what this actually indicates. In F1E why not show PAD1 as well as PAD2. In the text states PAD family members plural but only one example shown. No EP procyclin shown. Overall, while I think the presentation needs some work, otherwise all we have is that stumpy look like stumpy here, albeit with some lacuna in the data presented.

FS4 - This is very difficult to interpret. A whole cohort of genes are summed as evidence for differences in cell cycle-associated transcripts. But, there is clear evidence of differential expression for several of the slender forms, and the manner in which specific transcripts are weighted is not clear (I assume these have not been, but is that justified?). It is also not clear how transcripts have been selected - is this by knowledge, GO term or some other method. The problem here is that there is so much data compressed into a simple figure that it is impossible to evaluate these data in terms of support or otherwise for the concept being proposed, i.e. that cohort 1 has ceased division. For example, I do not agree that the data as presented do demonstrate that cluster 1 is non-dividing. I'd accept that it is less than elsewhere, but not for all cells. Which I would expect.

F2. The proportion of cells in panel B seems at odds with the transcriptomes for F1C? Such huge difference between animal 1 and 2 as is very clear from F2D makes me ask - how do we know what is representative here? I appreciate the costs involved but if we were to roll the die again, would we get a different picture? If that were true, how useful are the observations here?

Where are the data referred to on I196?

F3 - This is supposed to demonstrate the presence of differentiating forms I think, but the figure and text then describe only 2 of over 800 parasites with PAD1, i.e. less than 0.5% for the established marker. But then the flagellum is used here to justify this is differentiation, so I am left with the feeling that the author's would find evidence for differentiation regardless. Yet, there are clear differences between the cow and murine forms so how do we know what is actually happening here? My apologies, but this is very confusing,

F4 and 5. I find again that I really cannot follow the argument here. My big concern is that there is some cherry picking to find a story, when all that we seem to have is that transcriptomes in mice and cattle models are very different. Changes to ESAG expression and also H2A. I'm also unclear what is actually plotted in F4B, C, D, E and F. Specifically which time points??

SF6 - How are the cell types in C related to data in F2? What is a slender type 1 or type 2 or for that matter the stumpy forms?

Reviewer #3

(Remarks to the Author)

This study investigates the developmental dynamics of *Trypanosoma brucei* in cattle bloodstream infections. It describes how *T. brucei* population in cow blood comprises both slender and stumpy-like forms, supported by transcriptomic data, kinetoplast-nucleus (KN) configuration, and flagellum length measurements. Importantly, in cows, stumpy forms differ from those in mice, including PAD1 expression across the entire parasite surface. Additionally, the pseudo-bulk transcriptome of parasites in cows differs significantly from that of mice and in vitro samples, primarily due to differences in the stumpy-like population. Though largely descriptive, this study is highly valuable as it was conducted under veterinary-relevant conditions.

MAJOR CONCERNS

As mentioned in the discussion, a key aspect that remains to be addressed is the function of the stumpy-like forms found in cow bloodstream. Do they eventually differentiate into mature stumpy forms, meaning they are intermediate forms? Also, are stumpy-like forms as transmissible as fully mature stumpy forms? In other words, do they differ in the capacity to differentiate to procyclic forms? While these questions could be easily tested using well-established in vitro methods (differentiation to stumpy forms and differentiation to procyclic forms), they would require having access to parasites isolated from cows, which could be a challenge, unless authors have frozen vials.

In the PCA plot (Figure 4A), parasites from different hosts are clearly separated. However, these samples were sequenced separately at different time points. Were mice and cows infected from the same stock of vials? How can the authors be certain that the differences observed in Figure 4 are not due to batch effects? To strengthen the conclusions, the authors could include additional published single-cell datasets from both in vitro and in vivo settings in the PCA analysis. This would help distinguish real host-specific transcriptomic differences from potential technical artifacts.

The authors argue that the cow host environment drives transcriptomic adaptation and differentiation at lower parasite densities. This hypothesis could be experimentally tested. If the authors were to add blood or plasma from a cow—either non-infected or collected at different time points from infected animals—to parasites growing in vitro or freshly isolated from a mouse, would those parasites adopt a cow-specific transcriptomic profile, and how quickly would this transition occur? Furthermore, in this environment would slender forms begin differentiating into stumpy forms at lower densities? If differentiation is triggered, would it halt at the stumpy-like stage observed in cows, or would it progress to fully mature stumpy forms? These experiments would provide valuable insight into whether host-specific factors influence parasite development.

MINOR CONCERNS

Figure 1. Panels D and E are not cited in the text. At the same time, genes mentioned in the text are not shown in figures, such as genes associated to “glycolytic process” and “cytokinesis”.

Suppl. Figure 2E. Median number of genes/cells ranges between 400 and 800. In previous works, authors obtained ~1000 genes/cell. Could the authors comment on this difference?

The plot in Figure 2D presents three measurements from two samples using a bar graph with two pairs of columns, which is not intuitive. The reviewer suggests that the authors present the data in a table format instead. The table should classify each sample according to three key features: the percentage of cells in the 1K1N configuration, the percentage of cells in cluster C1, and the percentage of cells with a short flagellum. This format would provide a clearer and more interpretable representation of the relationships between these measurements.

On page 9, the statement 'No parasites showed expression of the PAD1 surface protein in either sample...' requires clarification. It would be important to highlight that the PAD1 staining shown in Figure 2A shows PAD1 expression at the flagellum, but not at the full surface of the parasite. To better appreciate the fact that stumpy forms in cows are different from stumpy forms in mice, authors should show PAD1-staining in stumpy forms from mice (expression at the surface) and slender forms from mice (no staining).

Figure 4. Given that HMI11 contains 10% FBS, wouldn't we potentially predict that transcriptomes of parasites in vitro and from a cow infection to be more similar than transcriptomes of parasites from a mouse infection? Could the authors comment on this point?

Suppl Figure 4. It is unclear why cell cycle analysis was inconclusive. Could the authors provide an explanation?

Figure 5. Given the reference mouse data is heavily biased toward stumpy forms, would this analysis be robust enough for the analysis of slender forms? On another note, if slender forms share similar transcriptomes between cow, mice and in vitro (Panel D), why did plating assays fail? Did authors try to inject parasites isolated from the cow into a mouse?

In Figure 4A, the PCA plot reveals a greater degree of variation among cow samples compared to mouse samples. Could authors generate bulk transcriptomes of only slender forms or stumpy-like clusters? This approach would enable confirmation of whether a particular life cycle stage contributes more significantly to the observed differences within cow samples or between hosts. Does Figure 5 suggest that stumpy-like forms contribute more significantly to the observed variability?

The authors state that integration of scRNA-seq data from mouse, cow, and in vitro samples was unsuccessful. Could they elaborate on the specific challenges encountered and provide possible explanations for this failure?

Reviewer #4

(Remarks to the Author)

Version 1:

Reviewer comments:

Reviewer #1

(Remarks to the Author)

The authors have carefully considered all my comments and addressed all my questions. As requested, the revision includes clear additional data and new aspects in the discussion.

Reviewer #2

(Remarks to the Author)

The authors have done an extensive rewrite and attempted to answer the issues raised by all reviewers. For the most part this is well done and improves the reading of the work. I remain unconvinced of major novelty here, but that is only an opinion and not a reason to delay publication further.

Reviewer #3

(Remarks to the Author)

The revised manuscript includes an additional experiment (Supplementary Fig. 8) showing that "adult cow serum alone was not sufficient to generate the cell types observed in samples from cattle bloodstream infection." While described in the Results section, this finding is not incorporated into the Discussion, which still states that "...gene expression changes, most likely related to adaptation to the host environment."

Supplementary Figures 5 and 6 show that anti-PAD1 serum produces two IFA patterns: flagellar and cell body surface signals. The authors interpret the flagellar signal as an artifact, as it is also seen in Lister 427 + BHI and AnTat slender forms from blood. Clarification is needed on whether "artifact" refers to cross-reactivity with a protein other than PAD1, or whether PAD1 in slender forms has a different localization and possibly a distinct function or inactive form. An antibody blocking experiment could help distinguish between these scenarios.

In the rebuttal, the authors provide a well-reasoned justification for not integrating transcriptomes from different hosts and demonstrate that label transfer, validated in Supplementary Figure 5, offers a more robust approach for this dataset. Given the relevance of this explanation to the interpretation of the analyses, it should be explicitly included in the manuscript rather than limited to the rebuttal.

A lower number of captured cells and reduced gene recovery per cell may be expected due to the extended processing time required for parasite isolation from cattle blood. However, as the version of the 10x Genomics kit can also influence the number of detected genes per cell, the manuscript should specify the exact kit version and chemistry used to generate the

transcriptomes.

Figure 7E–G show a positive correlation in fold-change values between datasets. However, the r values are not reported in the text, nor is there a biological interpretation of the differences between them. These plots appear disconnected from the accompanying text.

The legend of Supplementary Figure 5 should specify that the yellow arrows indicate slender forms.

Reviewer #4

(Remarks to the Author)

A REVIEWER COMMENTS

Reviewer #1 (Remarks to the Author):

In this nice manuscript, the authors have characterized the parasite populations that comprise acute and chronic phases of trypanosome infections in cattle and related these to established paradigms in mice and in vitro. They observed that infections are sustained at low parasitaemia and comprised of parasites that are not fully morphologically stumpy but express transcripts indicative of that developmental form. By establishing the molecular characteristics of parasites from the chronic phase of the infection, they have smartly extended beyond their previous rodent-based laboratory studies of acute infections and broadened analysis to parasites in chronic infections in the natural host. The discussion is very well written, putting the emphasis on the hypothesis drawn by this in-depth descriptive work. I only have very few comments.

We'd like to thank the reviewer for their interest in this study and feedback. We have addressed each concern raised below individually.

Main comment

In the title as well as along the entire manuscript, the authors assume that 'transmission-adapted bloodstream forms' are necessarily characterized by the expression of PAD1 over the entire plasma membrane. Is it correct? The authors state that 'No parasites showed expression of the PAD1 surface protein' on D11, however, figure 2 shows a clear PAD1 signal at the flagellum. In Larcombe et al. PNAS 2024, PAD1 seems to first be detected at the flagellum during the course of the differentiation. This is a pattern also frequently seen by IFA on AnTat1.1E BSF from mice (Calvo Alvarez et al. PLoS Path 2023). In total, the so-called early ST-like forms described in the manuscript present all characteristics of ST forms but a PAD1 expression restricted to the flagellum. Couldn't it be sufficient for progressing through the parasite cycle after transmission? In that case, transmissible forms wouldn't be so cryptic.

We thank the reviewer for the opportunity to clarify. In our hands, we consistently observe flagella staining with PAD1 antibody, including in non-differentiating proliferative slender populations. We therefore believe this to be an artefact (also seen with many other antibodies), distinct from the cell body surface coat staining of stumpy forms that express the PAD1 protein. For this reason, in both our previous study (Larcombe et al PNAS, DOI: 10.1073/pnas.2306848120) and the present study we classified parasites as PAD1 positive only when the signal was not restricted to the flagella or flagellar pocket.

To address this further we have performed an additional experiment, now presented in new Supplementary figure 6. Here we assess flagella PAD1 staining in both dividing, non-differentiating AnTat 1.1 90:13 pleomorphic parasites and monomorphic Lister 427 parasites, that are refractory to differentiation. This was done with and without brain-heart infusion (BHI) broth, known to trigger slender to stumpy differentiation in pleomorphic parasites (Rojas et al. (2019) Cell, DOI: 10.1016/j.cell.2018.10.041).

We demonstrate using identical sample preparation, imaging and processing that these monomorphic parasites also show a flagellum signal, and only by inducing differentiation in the pleomorphic AnTat strain do we see cell body surface staining with the PAD1 antiserum. The monomorphic Lister 427 parasites never have surface staining but always have flagella staining. Given that this strain is known to be unable to infect tsetse flies, we conclude flagella staining is not a marker of transmission-adaption.

We therefore do not believe flagellum fluorescence after PAD1 staining to be indicative of differentiation status in this study. Instead, this staining appears to be an artefact of the antiserum and does not indicate expression of functional PAD1 protein expression localised to the flagella.

We have clarified this now on lines 243-245 and with the addition of new supplementary figures 5 and 6.

Minor comments

Introduction

- L63-64: This is correct but it was not demonstrated for the first time in the cited reference.

Thank you for identifying this omission. We have added the additional reference of Herder et al 2007.

- The preponderance of extravascular parasites during an infection in the mammalian host could be already explained in the introduction as it is a major biological trait of Tbb biology.

We have now added this to the introduction (lines 82-88).

Results

- Figure 1: Could the authors show the PAD1 expression levels in panel E? Panel F apparently shows subtle differences that are difficult to detect.

We have now updated the figures in line with reviewer recommendations. The expression of individual members of the PAD family is now shown in greater detail in new figure 3 and supplementary figure 5.

- Were Factor H Receptor transcripts also enriched in Cluster 1?

Thank you for this suggestion. Interestingly, Tb927.5.4020 that encodes a Factor H receptor (Macleod et al 2020 Nature communications, 10.1038/s41467-020-15125-y) is significantly upregulated in cluster 1 as well, again indicating preadaptation.

We have added this gene to figure 3 and highlighted this in the text (line 196)

- Considering their surprising profile (upregulated TCA and pyruvate metabolism transcripts), what could be the nature / function of cells in cluster 3?

This is an interesting question and one we have explored in more detail in the updated manuscript, see new Figure 2. Parasites in cluster 3 were characterised by upregulation of metabolic genes, including succinyl-CoA synthetase (SCS), glycerol kinase (GK), oxoglutarate dehydrogenase (OGDH) and pyruvate dehydrogenase (PDH).

Taleva and colleagues (doi.org/10.1371/journal.ppat.1011699) recently showed that mitochondrial substrate-level phosphorylation via SCS is essential under *in vivo* conditions. This pathway provides BSF *T. brucei* with mitochondrial ATP via the conversion of succinyl-CoA to succinate. One potential source of this succinyl-CoA is acetyl-CoA that is generated by PDH (the pyruvate deriving from glucose catabolism). Another source is via the conversion of 2-oxoglutarate to succinyl-CoA (catalysed by OGDH). In this case, the 2-oxoglutarate would be derived from glutamine (Creek et al; doi.org/10.1371/journal.ppat.1004689; Johnston et al; doi.org/10.1042/BSR20181601), although transcripts associated with the proteins that catalyse this conversion, glutamine synthetase and glutamate dehydrogenase, were not enriched in cluster 3 (see new Fig 2). It is therefore likely that this specific cluster contains cells that exhibit elevated mitochondrial ATP production.

However, this doesn't fully explain the upregulation of glycerol kinase. Instead, this protein may be involved in gluconeogenesis, as recently shown in BSF (Kovarova et al; doi.org/10.1371/journal.pntd.0012007) and PCF (Wagnies et al; doi.org/10.1371/journal.ppat.1007502) *T. brucei*. This is supported by the upregulation (although not significant) of fructose-1,6-bisphosphatase (enzyme 5, Fig 2C). It should be noted that glucose concentration in the ruminant bloodstream is reduced compared to that of non-ruminants (~2 mM in the former, compared to ~5 mM in the latter). Therefore, gluconeogenesis would provide the parasite an alternate carbon source (glycerol) during cattle infections.

We have added these hypotheses to the manuscript discuss (lines 558-568)

- L194: Do you mean 'the proportion of proliferative parasites'?

Yes, corrected

- As previously described, e.g. in Pyana et al. PLoS NTDs 2011, strain isolation in culture or adaptation in laboratory mice are not trivial. In Van Reet et al. exp parasitol 2011, the authors used methylcellulose and human serum to support the continuous *in vitro* propagation of the bloodstream form. KIVI could also be useful to simultaneously assess the presence of parasites and their ability to differentiate into PCF *in vitro* (Truc P et al. Trans R Soc Trop Med Hyg 1992).

Thank you for the ideas. Unfortunately, the extremely low levels of parasites recovered from cattle infection prevent us testing varied culture conditions in the *ex vivo* replating assay and, similarly, the parasite density is too low to attempt *in vitro* differentiation to procyclic forms in an easily quantifiable assay.

Having previously, and easily, adapted the same Antat 1.1. 90:13 parasite line to culture from mouse infections successfully (Larcombe et al PNAS, DOI: 10.1073/pnas.2306848120), we do believe this failure to adapt parasites into culture from cattle was a specific effect related to parasites isolated from bovine hosts.

- Could PAD4, PAD6 and PAD7 ensure a redundant function in the differentiation process when PAD1 is barely expressed?

This is an interesting idea that we also considered. Please note that in updating our analyses for our revised manuscript we have presented the upregulation of more copies in the PAD array (see revised figure 3). However, previous transcriptome analysis has shown that only PAD1 and PAD2 are strongly elevated in stumpy forms (although PAD3 shows slight upregulation). Other PAD family members are largely restricted to expression in the tsetse midgut (e.g. PAD3, PAD5 and PAD 7) or salivary gland metacyclic forms (PAD 4 and 6) and so are unlikely to provide redundant function in differentiation (Naguleswaran et al., Plos Pathogens 2021, DOI: 10.1371/journal.ppat.1009239). We have added text to the discussion to highlight that the implications of a broad increase in PAD transcript levels we observed in the culture 1 population is currently unclear (Lines 520-524).

Discussion

- The authors fairly mention extravascular parasites in this section, but as all experiments were performed on parasites from the major venous circulation, could parasites in the micro-circulation be also different in terms of proliferation and transmissibility?

Yes, we agree this is possible and have added lines 503-505 in the discussion.

- L445: “The absence of fully morphologically stumpy forms in the cattle blood may be contributed to by an inability to successfully enter the circulatory system from the tissues, or by a more limited progression through the hierarchy of events that generate classic stumpy forms.” Again, why would it be necessary for parasites to reach the fully morphologically stumpy forms if a ST-like form (PAD1 at the flagellum only) is sufficient for transmission?

We have corrected this sentence to rectify this impression. It may be possible that the stumpy-like forms are sufficient for transmission, but this remains challenging to test with the current samples. We detail earlier why PAD1 staining at the flagellum is likely to be an antibody artefact and not representative of PAD protein expression.

NB: Some thoughts... Could the flagellum possibly be the first region of the cell surface where key proteins such as PAD1, and possibly also VSGs (e.g. during metacyclogenesis in Rotureau et al. Dev 2012) are de novo exported?

Please refer to our analysis of PAD1 staining above. This is an interesting idea, but again we do not have data to support this.

Reviewer #2 (Remarks to the Author):

Larcombe et al

This is another instalment in the studies by Briggs et al using scRNA-seq to characterise trypanosome differentiation. I think I have read most of these papers. This one looks at cattle infections using two animals and multiple time points, and which is a commendable goal as we have become aware of the limitations of small rodent models and their failure to accurately reflect infections in larger animals. The present work is technically sound, but my major issue is with presentation and interpretation. For a dataset such as this, providing a narrative is critical, as there are so many ways the data could be processed, and a justification really is needed for each analysis. Essentially, unpack, show some narrative and explain. I have some specific points that may help with moving the study forward. There does not appear to be a question, and there is no mechanistic insight and the abstract needs rewriting. I'd also argue that perhaps presenting the data in a more open minded fashion, with the obvious major thing being that the cow and murine models are hugely different, would make for a better story as well as perhaps a more impactful finding.

We'd like to thank the reviewer for their time in reviewing the manuscript, their feedback and interest our central goal.

Please see responses to their specific queries below. In summary we have made the following changes:

- Altered the abstract text as requested
- Altered figure 1, added revised figures 2, 3 and supplementary figure 4 to break down our investigation of the identified clusters more clearly.
- Substantially altered the text of the results section to explain the narrative.
- Added text to state the research questions explicitly. We feel that both the difference between cow and murine models, and the life cycle associated characteristics of these bloodstream forms to be significant findings that each exemplify the overall novelty and interest of our study.

L22 Probably need to be more specific on subspecies here.

We have edited the text to reflect that nearly all *T. brucei* subspecies (including *T. b. rhodesiense* and *T. b. gambiense* group 2) infect diverse non-human mammals, with the exception being *T. b. gambiense* group 1.

“The dominant hosts of *Trypanosoma brucei sensu lato* in tsetse fly endemic regions are non-human mammals, including agriculturally important cattle”. (lines 22-23)

L34 Obviously a key observation, but how is it that we know parasites are not replicating? Deserves a brief justification.

We have changed this to “detectably dividing” to reflect the lower proportion of 1N2K and 2N2K parasites in these samples, rather than referring to replicating.

I find the definition of differentiation extremely vague here - on the one hand we lack morphological or marker protein expression and on the other we have a shortened flagellum. So, does this mean that the shorter flagellum is NOT a differentiation marker?

The shorter flagellum is a documented marker for differentiation (Tyler, Matthews and Gull, Protist 2021, DOI: 10.1078/1434-4610-00074) in mouse models and in culture. In this work shorter flagella were evident in both intermediate forms and stumpy forms (identified by DHLADH expression).

Although, inevitably, the flagellum length is not a binary marker (unlike, for example PAD expression), we find a shorter flagellum to be associated with the increased proportion of parasites expressing stumpy associated transcripts. This indicates shorter flagella are also a marker in cattle samples, albeit not explicitly linked to PAD1 expression or “classical” stumpy morphology, as also observed in the original publication.

We have added this reference and the marker details earlier in the text to lines 241 - 242.

The idea that the transcriptome reflects the overall state of the cell seems flawed here as the proteome is the real thing. Perhaps the transcriptome per se is not a fair reflection of the cell's status.

We have reviewed the manuscript to correct this impression as we agree the transcriptome is not likely to directly reflect the proteome. Instead, we demonstrate the upregulation of stumpy-associated transcripts in a subpopulation of parasites, that we consider indicates the beginnings of adaptation. We have broken down our first results section to present this- please see revised figures 1-3 and the associated text with this aim in mind.

We have multiple terms here - stumpy, stumpy-like, extreme stumpy.... None of which have a precise delineation. This leads to some confusion as well as a somewhat uncomfortable sense that maybe even the author's are unclear on this. I appreciate we are not dealing with binary events here and that diversity is a major feature of biology, but the definition of terms is critical.

Like all developmental events, there are progressive steps between one cell type and the next. Our analyses will help to delineate this journey in molecular terms since in future work marker genes can be tagged to identify individual cells undergoing development. But in our revised text we have been more careful to justify or avoid terms such as “stumpy-like” or “extreme stumpy”, and have explicitly referred to the transcriptome in the summary and introduction sections of the manuscript. For example, we have also added a clarification on lines 200-201 stating that we refer to cluster 1 as ‘stumpy-like’ due to its transcriptomic signature.

F1 - Why number 0 to 3 and not 1 to 4?

This is the standard output of the computational package used for single cell transcriptomic analysis (widely used across the literature). We prefer not to alter this as it does not change the conclusions.

Reason for big difference between cow 1 and 2 on day 11, and to lesser extent elsewhere?

This inconsistency between animals is not unusual in experimental cattle experiments, with animals being outbred and having experienced different environmental exposure prior to sourcing from farm stocks. Indeed, the parasitaemia difference over time and between cows in our studies are similar to earlier experimental infections with the same parasite strain in our research facility (Saldanha et al 2024, Scientific Reports, DOI: 10.1038/s41598-024-54857-5). As in our studies, the clearest difference between calves was early in infection (before around day 20), with parasitaemia generally persisting at lower levels after this point. In the Saldanha *et al* study, the day on which parasitaemia peaked also varied between animals. But, critically, our studies focus on the 'within host' rather than 'between host' temporal development and transcriptome of the parasites in cattle such that differences between animals are less relevant, although are often informative.

We have added a sentence to the manuscript (line 130-131) to highlight that these patterns are consistent with other published work.

Panel D not aligned and based on the text is unclear what this actually indicates.

We have altered the relevant figure legend and methods (lines 647-649) to explain that this is the average expression of the previously identified marker genes per cell. The plot shows the average expression of marker genes for each cell grouped by cluster.

In F1E why not show PAD1 as well as PAD2. In the text states PAD family members plural but only one example shown. No EP procyclin shown. Overall, while I think the presentation needs some work, otherwise all we have is that stumpy look like stumpy here, albeit with some lacuna in the data presented.

This figure presented the top marker genes per cluster. We have replaced it with a dot plot to present more markers for each cluster (Figure 1E). These are the top markers in terms of fold-change and are shown to avoid any bias in marker presentation.

We have presented the expression of the PADs specifically in revised figure 3B, including PAD1.

FS4 - This is very difficult to interpret. A whole cohort of genes are summed as evidence for differences in cell cycle-associated transcripts. But, there is clear evidence of differential expression for several of the slender forms, and the manner in which specific transcripts are weighted is not clear (I assume these have not been, but is that justified?).

It is also not clear how transcripts have been selected - is this by knowledge, GO term or some other method. The problem here is that there is so much data compressed into a simple figure that it is impossible to evaluate these data in terms of support or otherwise for the concept being proposed, i.e. that cohort 1 has ceased division. For example, I do not agree that the data as presented do demonstrate that cluster 1 is non-dividing. I'd accept that it is less than elsewhere, but not for all cells. Which I would expect.

We have updated our cell cycle analysis, now presented in Figure 1E and supplementary figure 4.

The transcripts used for phase markers are from our previous studying of the cell cycle (Briggs et al *Elife* 2023, DOI: 10.7554/eLife.86325) using proliferating bloodstream forms. The markers were filtered to use those detected in 10% or more of the cells (1 or more transcript/UMI), to remove those detected very sparsely in these data.

We then used this set of genes to calculate the average expression score for each phase. This is the average expression of the marker gene list per cell, minus the average expression of a control set of randomly selected genes. This is implemented with Seurat tool "AddModuleScore" and the scores are presented in new supplementary figure 4A.

For labelling each cell with a phase, if the average expression of all phases (G1, S and G2/M) was not increased over the control gene set (i.e. all scores <0), this is labelled as "Unlabelled". These are likely to be non-dividing when captured as they do not express transcripts we know are enriched in G1, S and G2/M phases.

We have clarified the new approach in the text (lines, 156-161 and 646-649). And new figures are now in Figures 1F and supplementary figure 4.

F2. The proportion of cells in panel B seems at odds with the transcriptomes for F1C? Such huge difference between animal 1 and 2 as is very clear from F2D makes me ask - how do we know what is representative here? I appreciate the costs involved but if we were to roll the die again, would we get a different picture? If that were true, how useful are the observations here?

In our new cell cycle analysis, it is clear that clusters 0 and 3 also have large proportions of G1 or unlabelled parasites (figure 1E and supplementary figure 4B), and so an exact match between cluster proportion and KN proportion is not expected.

However, when comparing the proportions of 1N2K and 2N2K parasites in each sample (now Figure 4B) with the proportions of S and G2/M labelled parasites using transcriptomic signatures (Figure 1F), there is a similar pattern as expected: day 4 for both cows have the largest proportions of these, and these are reduced in days 11 and day 23.

A bigger sample size of animals may be useful (though prohibitively expensive!) but given that we observed similar bloodstream form subpopulations (i.e. clusters) in multiple samples and across both infections, this indicates consistency in the parasites across the infection profiles. As discussed above differences between individual cattle are expected in our model and the significant comparison is the temporal pattern within an animal, and not differences between animals.

Where are the data referred to on I196?

More detailed results of this experiment have been added to the results text (lines 227-234) as well as detailed methods section (lines 683-699).

F3 - This is supposed to demonstrate the presence of differentiating forms I think, but the figure and text then describe only 2 of over 800 parasites with PAD1, i.e. less than 0.5% for the established marker. But then the flagellum is used here to justify this is differentiation, so I am left with the feeling that the author's would find evidence for differentiation regardless. Yet, there are clear differences between the cow and murine forms so how do we know what is actually happening here? My apologies, but this is very confusing,

Please see our response above regarding the flagellum length as a marker of the transition between slender, intermediate and stumpy forms. The data demonstrates that although PAD expression is rare, there is clear evidence that the parasite populations in cattle are not equivalent to replicative slender forms (i.e. matching parasites at d3 in mouse infections). Nor are they stumpy (equivalent to PAD1+ parasites at day 6-7 in mice). Instead, they are variably distributed (in terms of flagellar length) between these extremes, being most similar to parasites at day 5 of a mouse infection (intermediate forms). This intermediate flagellar length is maintained throughout the chronic phase of infection (Figure 5). These chronic stage parasites, derived from cattle, have never been morphologically analysed previously.

We have amended the figure legend title, section title (lines 271 – 272) and results text (lines 293-294) to clarify that these results indicate mixed populations in terms of development.

F4 and 5. I find again that I really cannot follow the argument here. My big concern is that there is some cherry picking to find a story, when all that we seem to have is that transcriptomes in mice and cattle models are very different. Changes to ESAG expression and also H2A . I'm also unclear what is actually plotted in F4B, C, D, E and F. Specifically which time points??

We have amended the text (Lines 313-16) and figure legend (1008-1009) to clarify that in Figures 4B-F, all time points are used for analysis for each condition.

We carried out these differential expression tests to look for broad host specific changes. In volcano plots, the top differentially expressed genes with largest fold changes are labelled unbiasedly, and all significant genes are used for GO term analysis to summarise the findings, without cherry picking. As the top GO terms linked to the cattle host environment (i.e. "purine containing compound salvage")

were largely driven by the presence of adenylate cyclases, we investigated these further in the figure G.

All differential expression results and GO terms are available in supplementary data 2.

SF6 - How are the cell types in C related to data in F2? What is a slender type 1 or type 2 or for that matter the stumpy forms?

We have now altered this analysis and used broader slender and stumpy labels to avoid confusion. This is presented in new simplified figure 7.

Reviewer #3 (Remarks to the Author):

This study investigates the developmental dynamics of *Trypanosoma brucei* in cattle bloodstream infections. It describes how *T. brucei* population in cow blood comprises both slender and stumpy-like forms, supported by transcriptomic data, kinetoplast-nucleus (KN) configuration, and flagellum length measurements. Importantly, in cows, stumpy forms differ from those in mice, including PAD1 expression across the entire parasite surface. Additionally, the pseudo-bulk transcriptome of parasites in cows differs significantly from that of mice and in vitro samples, primarily due to differences in the stumpy-like population. Though largely descriptive, this study is highly valuable as it was conducted under veterinary-relevant conditions.

We thank the reviewer for their time in considering this work, and for providing feedback and ideas. We have addressed the below queries individually.

MAJOR CONCERNS

As mentioned in the discussion, a key aspect that remains to be addressed is the function of the stumpy-like forms found in cow bloodstream. Do they eventually differentiate into mature stumpy forms, meaning they are intermediate forms? Also, are stumpy-like forms as transmissible as fully mature stumpy forms? In other words, do they differ in the capacity to differentiate to procyclic forms? While these questions could be easily tested using well-established in vitro methods (differentiation to stumpy forms and differentiation to procyclic forms), they would require having access to parasites isolated from cows, which could be a challenge, unless authors have frozen vials.

These are interesting questions we have also discussed. Unfortunately, as the reviewer highlights, the capacity to perform such experiments with parasites directly isolated from cattle is limited by the availability of samples and the low parasite density, which limits a quantifiable analysis of onward differentiation.

We however thank the reviewer for the suggested in vitro experiments, which we have attempted and discuss the results of below.

In the PCA plot (Figure 4A), parasites from different hosts are clearly separated. However, these samples were sequenced separately at different time points. Were mice and cows infected from the same stock of vials? How can the authors be certain that the differences observed in Figure 4 are not due to batch effects?

Cattle infections were started from the blood of an infected donor mouse. This mouse infection was started from the same strain as previous mouse infections in the PCA, however the mouse infections used a derivative of this strain (AnTat 1.1 90:13) that carried RNAi targeting fragment relevant to the previously study (Larcombe et al PNAS, DOI: 10.1073/pnas.2306848120). The BHI treated *in vitro* samples included in this comparison also used the same strain, AnTat 1.1 90:13.

Thank you for highlighting the potential batch affects. Unfortunately, we cannot specifically control for these as each experiment was performed independently and so the condition and batch cannot be explicitly separated. This was an inevitable consequence of the practical restrictions of deriving material from cattle in a contained large animal facility (geographically isolated from the mouse facility) and the practical restriction of isolating these parasites for analysis. Despite this limitation, we feel this analysis is valuable given that no such comparisons have been possible previously. Their inclusion is also supported by the close relationship between the *in vitro* and mouse sample analyses, despite these samples being processed independently. Of note, cow samples analysed from the same batch (i.e. day 4, day 11 or day 23 batches), do not always group together. Hence samples in the PCA plot are not grouped solely by experimental batch and biologically meaningful variation is present.

To strengthen the conclusions, the authors could include additional published single-cell datasets from both *in vitro* and *in vivo* settings in the PCA analysis. This would help distinguish real host-specific transcriptomic differences from potential technical artifacts.

We are not sure which other single cell datasets the reviewer is referring to. The data in the current PCA plot are derived from *in vitro* single cell analysis of bloodstream forms (Briggs et al Nature Comms, DOI: 10.1038/s41467-021-25607-2.) and from mouse derived single cell samples (Larcombe et al PNAS, DOI: 10.1073/pnas.2306848120), which we believe are the only published analyses of pleomorphic bloodstream trypanosomes. Other published *T. brucei* single cell data are derived from tsetse fly stages and so are not included nor relevant here. We have altered the text to make this clear earlier in the results (lines 303-304) and figure legend (lines 1004-1005).

The authors argue that the cow host environment drives transcriptomic adaptation and differentiation at lower parasite densities. This hypothesis could be experimentally tested. If the authors were to add blood or plasma from a cow—either non-infected or collected at different time points from infected animals—to parasites growing *in vitro* or freshly isolated from a mouse, would those parasites adopt a cow-specific transcriptomic profile, and how quickly would this transition occur? Furthermore, in this environment would slender forms begin differentiating into stumpy forms at lower densities? If differentiation is triggered, would it halt at the stumpy-like stage observed in cows, or would it progress to fully mature stumpy forms? These experiments would provide valuable insight into whether host-specific factors influence parasite development.

Thank you for the suggestions.

Although transcriptomic analysis was not feasible, we performed the suggested experiment to test if parasites cultured in the presence of cow serum/plasma exhibit changes in their KN counts, PAD1 expression or flagella length, reflecting what is seen in cattle infections. The experiments were limited because cultured bloodstream forms quickly die when exposed to high concentrations of adult cow serum. However, by using 10% adult cow serum we found the parasites could adapt and tolerate adult bovine serum. In these conditions, growth was unaffected and the parasites did not show any evidence of accelerated development to stumpy forms. Within the limitations of the experiment this indicates that adult cow serum alone was not sufficient to generate the cell types we observed in samples isolated from the cattle bloodstream infection.

The results of these experiments are presented in new supplementary figure 8 and results text (lines 431-441).

MINOR CONCERNS

Figure 1. Panels D and E are not cited in the text. At the same time, genes mentioned in the text are not shown in figures, such as genes associated to “glycolytic process” and “cytokinesis”.

Apologies-this was an oversight. We have since made changes to these figures and text.

Suppl. Figure 2E. Median number of genes/cells ranges between 400 and 800. In previous works, authors obtained ~1000 genes/cell. Could the authors comment on this difference?

Although we cannot point to a specific step that has reduced the recovery of genes per cell in these experiments, since this was consistent at each time point, we think this difference is likely due to impact of the longer processing time required to obtain parasites from cattle blood rather than subsequent steps. This involved unavoidable lysis of the red blood cells in large volumes, multiple centrifugation steps as well as large scale column purification.

The plot in Figure 2D presents three measurements from two samples using a bar graph with two pairs of columns, which is not intuitive. The reviewer suggests that the authors present the data in a table format instead. The table should classify each sample according to three key features: the percentage of cells in the 1K1N configuration, the percentage of cells in cluster C1, and the percentage of cells with a short flagellum. This format would provide a clearer and more interpretable representation of the relationships between these measurements.

Thank you for this suggestion. We have now presented these data as a table as suggested.

On page 9, the statement 'No parasites showed expression of the PAD1 surface

protein in either sample...' requires clarification. It would be important to highlight that the PAD1 staining shown in Figure 2A shows PAD1 expression at the flagellum, but not at the full surface of the parasite. To better appreciate the fact that stumpy forms in cows are different from stumpy forms in mice, authors should show PAD1-staining in stumpy forms from mice (expression at the surface) and slender forms from mice (no staining).

We have added PAD1 staining for mouse samples as suggested to supplementary figure 5 and amended the relevant text. As highlighted in response to Referee 1 we have now confirmed that flagellar staining does not represent bona fide and functional PAD1 expression.

Figure 4. Given that HMI11 contains 10% FBS, wouldn't we potentially predict that transcriptomes of parasites *in vitro* and from a cow infection to be more similar than transcriptomes of parasites from a mouse infection? Could the authors comment this point?

In fact, the *in vitro* cultured parasites used in this study were cultured in 20% foetal calf serum. Although we agree this might have been anticipated to make the *in vitro* and cattle derived samples more similar, this was not our observation. This might reflect the substantial differences that exist between foetal and adult bovine serum, also reflected in the inability of cultured parasites to grow in high concentrations of adult bovine serum.

Suppl Figure 4. It is unclear why cell cycle analysis was inconclusive. Could the authors provide an explanation?

We have now updated this approach to avoid an arbitrary threshold of phase labelling, which has improved this analysis. Please see our response to reviewer 2 and revised figure 1F and supplementary figure 4.

Figure 5. Given the reference mouse data is heavily biased toward stumpy forms, would this analysis be robust enough for the analysis of slender forms? On another note, if slender forms share similar transcriptomes between cow, mice and *in vitro* (Panel D), why did plating assays fail?

Please see new figure 7. Here we see that the analyses of slender and stumpy-like forms from the mouse correlates well with the equivalent *in vitro* dataset (which has a far higher proportion of slender forms).

In reference to the replating assay, it is possible that the necessary adaption for growth in cattle serum occurs at the proteome or metabolic level, rather than transcriptomic level. Alternatively, transcriptome differences not highlighted in our cluster markers might contribute.

Did authors try to inject parasites isolated from the cow into a mouse?

No, we did not attempt this experiment with these infections.

In Figure 4A, the PCA plot reveals a greater degree of variation among cow samples compared to mouse samples. Could authors generate bulk transcriptomes of only slender forms or stumpy-like clusters? This approach would enable confirmation of whether a particular life cycle stage contributes more significantly to the observed differences within cow samples or between hosts. Does Figure 5 suggest that stumpy-like forms contribute more significantly to the observed variability?

To generate selected transcriptomes of the slender and stumpy-like populations would require an experimental approach to sort these cell types which exist in a mixed population during infection. This would require extensive protein tagging of marker genes and then positive sorting of the cell types via FACS; clearly this is beyond the scope of the current study.

Yes, figure 5 does suggest that the stumpy like forms contribute more significantly to the variability. We have now amended these plots to clarify this in new figure 7.

The authors state that integration of scRNA-seq data from mouse, cow, and in vitro samples was unsuccessful. Could they elaborate on the specific challenges encountered and provide possible explanations for this failure?

We attempted integration with several tools (Seurat, Harmony and fastMNN) and each gave a different result, reducing our confidence in this analysis. For example, when using Seurat anchor-based integration, very few anchor points were found which may have been due to the lower number of genes/cells in the cattle data. We also found that the low proportion of slender forms in the mouse samples was not preserved after integration, again reducing our confidence in these approaches. Finally, we often found the markers for resulting clusters were not consistent for each dataset that had been integrated. This variability is well documented for different integration methods (see, DOIs: [10.1038/s41587-021-00895-7](https://doi.org/10.1038/s41587-021-00895-7) and [10.1038/s41592-021-01336-8](https://doi.org/10.1038/s41592-021-01336-8)) and in the absence of “ground-truth” to validate these approaches with our data, we chose to perform label transfer after validating this approach in supplementary figure 5.

Reviewer #4 (Remarks to the Author):

Reponses to reviewer comments

We'd like to thank all the reviewers for their time and efforts in reviewing this manuscript, and for their helpful suggestions. We have responded to the remaining queries below.

REVIEWERS' COMMENTS

Reviewer #1 (Remarks to the Author):

The authors have carefully considered all my comments and addressed all my questions. As requested, the revision includes clear additional data and new aspects in the discussion.

Reviewer #2 (Remarks to the Author):

The authors have done an extensive rewrite and attempted to answer the issues raised by all reviewers. For the most part this is well done and improves the reading of the work. I remain unconvinced of major novelty here, but that is only an opinion and not a reason to delay publication further.

Reviewer #3 (Remarks to the Author):

The revised manuscript includes an additional experiment (Supplementary Fig. 8) showing that “adult cow serum alone was not sufficient to generate the cell types observed in samples from cattle bloodstream infection.” While described in the Results section, this finding is not incorporated into the Discussion, which still states that “...gene expression changes, most likely related to adaptation to the host environment.”

To the discussion we have added:

“Notably, addition of adult cow serum to cultured parasites did not mirror this rapid *in vivo* adaptation, indicating more complex host-parasite interactions.”

Supplementary Figures 5 and 6 show that anti-PAD1 serum produces two IFA patterns: flagellar and cell body surface signals. The authors interpret the flagellar signal as an artifact, as it is also seen in Lister 427 + BHI and AnTat slender forms from blood. Clarification is needed on whether “artifact” refers to cross-reactivity with a protein other than PAD1, or whether PAD1 in slender forms has a different

localization and possibly a distinct function or inactive form. An antibody blocking experiment could help distinguish between these scenarios.

As previous work using anti-PAD1 (Dean S, et al . Nature. 2009 PMID: 19444208) in western blotting did not reveal PAD1 protein in slender forms, we believe the artifact due to the antibody binding to an alternative protein in the flagellum rather than alternative PAD1 localisation.

In the rebuttal, the authors provide a well-reasoned justification for not integrating transcriptomes from different hosts and demonstrate that label transfer, validated in Supplementary Figure 5, offers a more robust approach for this dataset. Given the relevance of this explanation to the interpretation of the analyses, it should be explicitly included in the manuscript rather than limited to the rebuttal.

We feel a detailed explanation is too length for the manuscript text, however we have made a minor modification to explain the absence of a means to validate the integration approaches.

“In the absence of “ground-truth” for these methods we instead used a label transfer...”

A lower number of captured cells and reduced gene recovery per cell may be expected due to the extended processing time required for parasite isolation from cattle blood. However, as the version of the 10x Genomics kit can also influence the number of detected genes per cell, the manuscript should specify the exact kit version and chemistry used to generate the transcriptomes.

This has now been added.

Figure 7E–G show a positive correlation in fold-change values between datasets. However, the r values are not reported in the text, nor is there a biological interpretation of the differences between them. These plots appear disconnected from the accompanying text.

Text has now been added:

“The highest correlation was evident between *in vitro* and mouse-derived datasets ($r = 0.73$, $p < 0.05$), with changes between slender and stumpy-like forms showing slightly lower correlation between these data and cattle samples (cow vs. mouse: $r = 0.59$, $p < 0.04$; cow vs. *in vitro*: $r = 0.68$, $p < 0.05$).”

The legend of Supplementary Figure 5 should specify that the yellow arrows indicate slender forms.

Added